# Decoding the genetic structure of conjugative plasmids in international clones of *Klebsiella pneumoniae*: A deep dive into *bla*$_{KPC}$, *bla*$_{NDM}$, *bla*$_{OXA-48}$, and *bla*$_{GES}$ genes

**Shadi Aghamohammad**[1☯]**, Mahshid Khazani Asforooshani**[1,2☯]**, Yeganeh Malek Mohammadi**[1]**, Mohammad Sholeh**[1]**, Farzad Badmasti**[1] *

**1** Department of Bacteriology, Pasteur Institute of Iran, Tehran, Iran, **2** Department of Microbiology, Faculty of Biological Sciences, Alzahra University, Tehran, Iran

☯ These authors contributed equally to this work.
* fbadmasti2008@gmail.com

## Abstract

Carbapanem-resistant *Klebsiella pneumoniae* is a globally healthcare crisis. The distribution of plasmids carrying carbapenemase genes among *K. pneumoniae* poses a serious threat in clinical settings. Here, we characterized the genetic structure of plasmids harboring major carbapenemases (e.g. *bla*$_{KPC}$, *bla*$_{NDM}$, *bla*$_{OXA-48}$-like, and *bla*$_{GES}$) from *K. pneumoniae* using bioinformatics tools. The plasmids carrying at least one major carbapenemase gene were retrieved from the GenBank database. The DNA length, Inc type, and conjugal apparatus of these plasmids were detected. Additionally, allele types, co-existence, co-occurrence of carbapenemase genes, gene repetition, and sequence types of isolates, were characterized. There were 2254 plasmids harboring carbapenemase genes in the database. This study revealed that *bla*$_{KPC-2}$, *bla*$_{NDM-1}$, *bla*$_{OXA-48}$, and *bla*$_{GES-5}$ were the most prevalent allele types. Out of 1140 (50%) plasmids were potentially conjugative. IncFII, IncR, IncX3, and IncL replicon types were predominant. The co-existence analysis revealed that the most prevalent of other resistance genes were *bla*$_{TEM-1}$ (related to *bla*$_{KPC}$), *bla*$_{OXA-232}$ (related to *bla*$_{OXA-48}$), *ble*$_{MBL}$ (related to *bla*$_{NDM}$), and *aac (6')-Ib4* (related to *bla*$_{GES}$). The co-occurrence of carbapenemases was detected in 42 plasmids while 15 plasmids contained carbapenemase gene repetitions. Sequence alignments highlighted that plasmids carrying *bla*$_{KPC}$ and *bla*$_{OXA-48}$-like were more homogeneous whereas the plasmids carrying *bla*$_{NDM}$ were divergent. It seems that *K. pneumoniae* utilizes diversity of genetic flexibility and recombination for resistance against carbapenems. The genetic structure of the plasmids showed that class I and III, Tn3 family, Tn5403 family derivatives, and Tn7-like elements were strongly associated with carbapenemases. The mobilizable plasmids carrying carbapenemases play an important role in the spread of these genes. In addition, gene repetition maybe is related to carbapenem heteroresistance. According to MST (minimum spanning tree) results, the majority of plasmids belonged to sequence type (ST) 11, ST14, and ST12. These international clones have a high capacity to acquire the carbapenemase-containing plasmids.

**Data Availability Statement:** The accession numbers of the datasets generated and analyzed during the current study are available in S4 File. All plasmid sequences can be retrieved from the GenBank database (https://www.ncbi.nlm.nih.gov/genbank/) or Batch Entrez (https://www.ncbi.nlm.nih.gov/sites/batchentrez).

**Funding:** The author(s) received no specific funding for this work.

**Competing interests:** The authors have declared that no competing interests exist.

## 1. Introduction

*Klebsiella pneumoniae* is one of the most important opportunistic pathogens found in nosocomial and community-acquired infections as well as in asymptomatic fecal carriages [1, 2]. *K. pneumoniae* isolates play an important role in causing serious infections, including pneumonia, bloodstream infections, urinary tract infections, surgical site, and burn wound infections [3]. In addition, the presence and spread of resistance genes pose a challenge to successful treatment. Extended-spectrum beta-lactamase (ESBL)-producing and carbapenemase-producing *K. pneumoniae* (CPK) isolates are repeatedly associated with failure of antibiotic therapy [4]. The mortality rate caused by carbapenem-resistant *K. pneumoniae* (CRK) isolates is significantly twice as high compared with infections caused by carbapenem-susceptible isolates [5]. Several reasons, including severe co-morbidities, higher virulence of CRK isolates, improper use of antibiotics along with high level of toxicity, are associated with the increase in mortality rates [6].

Various classes of carbapenemase, according to Ambler classification, are detected in *K. pneumoniae* isolates, including class B, metallo-beta-lactamases (New Delhi Metallo-beta-lactamase—NDM), class D carbapenemases (*e.g.* OXA-48), and class A carbapenemase of *K. pneumoniae* (*e.g.* KPC) [7]. Although CPK isolates are common in different regions, KPC is endemic in the United States and some European countries, including Greece and Italy. Whereas, MBLs (metallo-beta-lactamases, including NDM-1) and OXA-48-like are found mainly in Asian countries such as Turkey, India, Pakistan, and the Middle East region [8]. It seems that the presence of carbapenem resistance genes on conjugative plasmids could lead to their worldwide dissemination and therefore the high prevalence of CPK isolates could be a serious problem for the health care system all over the world. Also, the coexistence of carbapenemase-encoding genes with other resistance genes such as aminoglycoside-modifying genes in *K. pneumoniae* isolates exacerbates the problem of antibiotic resistance as a challenge in the curing of infectious [9, 10]. The harboring of these resistance genes on mobile genetic elements (MGEs), including class 1 integron, transposons, and insertion sequences are usually carried on conjugative plasmids, contribute to expansion of antimicrobial resistance (AMR).

Resistance genes are usually associated with specific clonal groups. Strains of multidrug-resistant *K. pneumoniae* isolates are generally found in sequence type (ST) 147, ST15, and ST258. In addition, virulence genes are also carried in several specific STs, including ST147, ST15, ST48, ST101, and ST383. Plasmids harboring carbapenem-resistance genes, including $bla_{KPC}$ belonging to IncFIIk/IncR, are typically found in ST11 and play an important role in the spread of resistance genes in Asian countries, including China [11]. Moreover, ST11 is highly related to hypermucoviscous isolates of *K. pneumoniae*. Isolates of carbapenem-resistant *K. pneumoniae* ST11 can acquire the virulence of large plasmids from hypervirulent isolates [12]. In other words, virulence plasmids from hypervirulent *K. pneumoniae* isolates can be transferred to resistant isolates [13] and Therefore, the spread of these high virulence and resistance capacity plasmids among prevalent STs such as ST11 could be complicated and potentially life-threatening for patients, especially those hospitalized in intensive care units (ICUs) for a long time [14].

According to resistant *K. pneumoniae*, various epidemiological studies have been conducted so far. However, it is still important to analysis that led to the decipher of the genetic structures of plasmids harboring AMR genes. Also the investigation of the MGEs associated with the resistant plasmids and prevalent STs, play an important role in the spread of antibiotic resistance among bacteria. Characterization of these MGEs can give rise to a better insight of how antimicrobial resistance might widely spread. In this study, we detected and compared the different allele types of the major carbapenemase genes, including $bla_{KPC}$, $bla_{NDM}$, $bla_{OXA-}$

48, and *bla*GES from *K. pneumoniae* using bioinformatics tools. In addition, we characterized the genetic properties of carbapenemases harboring plasmids including replicon types, conjugation ability, the co-existence (*e.g.* linkage of carbapenemases with other antimicrobial resistance genes), co-occurrence (*e.g.* having at least two carbapenemase genes in one strain), gene repetition, alignment and phylogenetic relatedness.

## 2. Materials and methods

### 2.1. Preparation of initial dataset

The complete nucleotide sequences of the plasmids containing each of the four carbapenemase genes, including *bla*KPC, *bla*NDM, *bla*OXA-48, and *bla*GES were retrieved from the GenBank database (https://www.ncbi.nlm.nih.gov/genbank/). To get all completed plasmids and partial DNA fragments carrying carbapenemase genes two types of BLASTn including microbial BLAST (https://blast.ncbi.nlm.nih.gov/Blast.cgi?PAGE_TYPE=BlastSearch&BLAST_SPEC= MicrobialGenomes) and standard BLAST (https://blast.ncbi.nlm.nih.gov/Blast.cgi? PROGRAM=blastn&BLAST_SPEC=GeoBlast&PAGE_TYPE=BlastSearch) were performed, respectively. Further analyses which have been conducted in the current study, are presented in Fig 1.

### 2.2. Allele types determination of carbapenemase genes

The allele types of the mentioned carbapenemase genes located on the completed plasmids and partial DNA fragments were detected using the beta-lactamase database (http://bldb.eu/). The criteria were 100% identity and 100% coverage. In addition, the prevalence of each allele type was calculated.

### 2.3. Detection of other AMR genes on retrieved DNAs

The Comprehensive Antibiotic Resistance Database (CARD) (https://card.mcmaster.ca/ home) was used to detect the presence of antimicrobial resistance genes against carbapenem, extended-spectrum beta-lactams, fluoroquinolosides, aminoglycosides, chloramphenicol, tetracycline, macrolide, and other antibiotics [15]. The co-existence (gene linkage) of other antimicrobial resistance genes with the major carbapenemase genes in the plasmids was characterized. Moreover, the availability of the at least two major carbapenemase genes in an isolate were considered as co-occurrence.

### 2.4. Genetic evaluation of plasmid harboring carbapenemase genes

The conjugal apparatus including *oriT*, relaxase, type IV coupling protein (T4CP), and type IV secretion system (T4SS) was detected by oriTfnder tool (https://tool-mml.sjtu.edu.cn/ oriTfinder/oriTfinder.html) [16]. The incompatibility (Inc) group of plasmids was identified by the Center for Genomic Epidemiology (CGE) web tool PlasmidFinder 2.1 (https://cge.food. dtu.dk/services/PlasmidFinder/) [17]. The ClustAGE software package (http://vfsmspineagent. fsm.northwestern.edu/cgi-bin/clustage_plot.cgi) was applied to compare the similarity/heterogeneity of plasmids carrying the predominant carbapenemase genes, including *bla*NDM-1, *bla*OXA-48-like, and *bla*KPC-2 and the results were depicted by EvolView (www.evolgenius.info/ evolview) [18]. In addition, the data on the geographical regions, isolation sources, years and hosts of all isolates harboring crabapenemase genes were extracted and summarized.

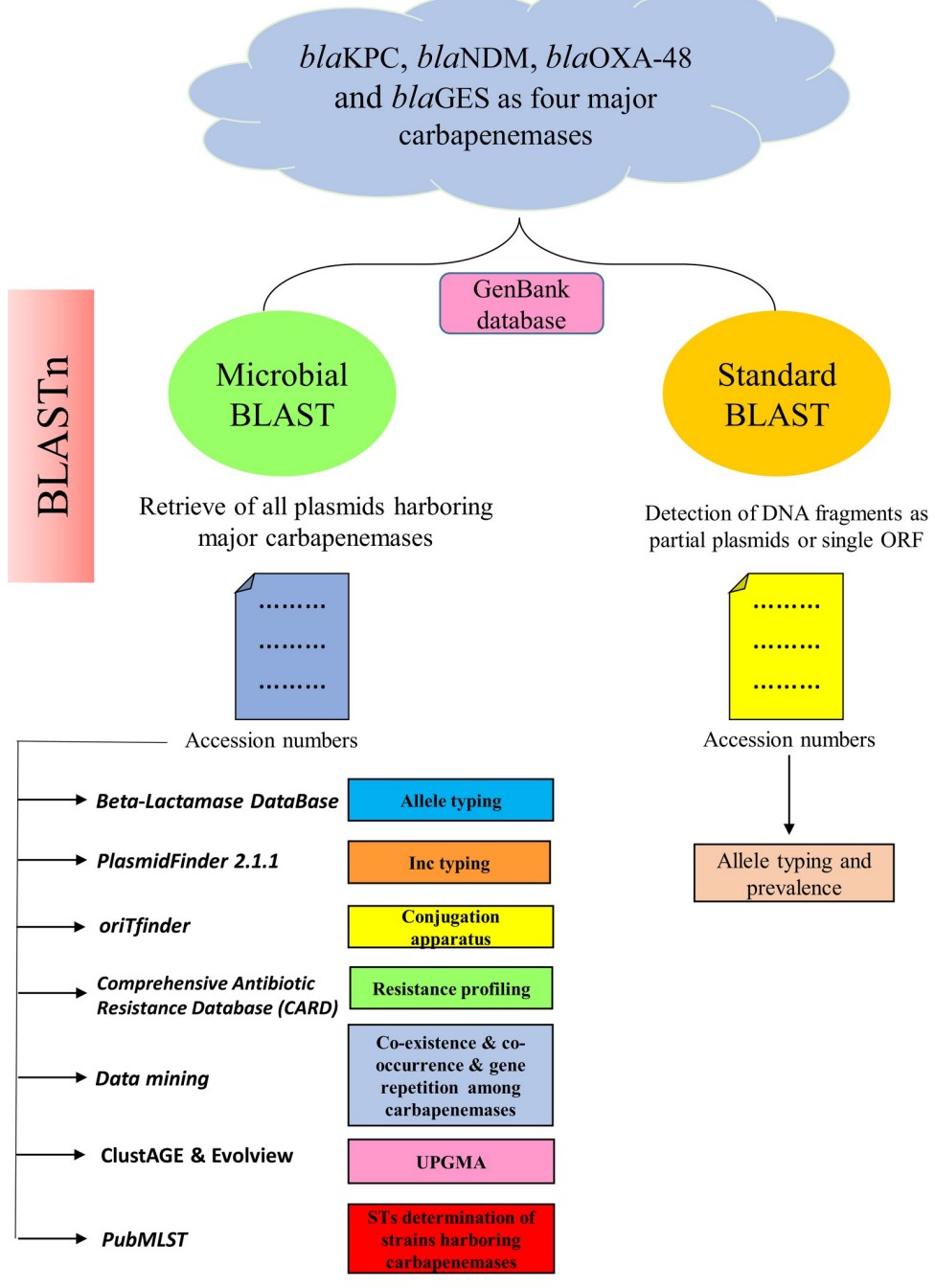

**Fig 1. The flowchart conducted in the current study.** Two BLAST approaches including microbial BLAST and standard BLAST had been applied to retrieve all completed plasmids and DNA fragments carrying carbapenemase genes. All tools and functions have been shown in this pipeline.

## 2.5. Clonal relatedness of strains harboring carbapenemase genes

The distribution of the major carbapenemase-encoding genes among the different STs was assessed. For each plasmid, the ST of the associated chromosome was determined using seven housekeeping genes (*gapA*, *infB*, *mdh*, *pgi*, *phoE*, *rpoB*, and *tonB*) via the PubMLST database

(https://pubmlst.org/) [19]. The clonal relatedness of STs was characterized using PHYLOViZ version 2.0 to generate a minimum spanning tree (MST) for all STs [20].

## 3. Results

### 3.1. The distributions and the allele types of the carbapenemase genes

Two thousand two hundred and fifty-four (2254), including 1132 plasmids harboring $bla_{KPC}$, 495 plasmids containing $bla_{NDM}$, 617 plasmids with $bla_{OXA-48}$-like, and 10 plasmids with $bla_{GES}$ were retrieved from GenBank database in microbial BLAST. In addition, 362, 132, 124, and 11 DNA partial fragments carrying $bla_{KPC}$, $bla_{NDM}$, $bla_{OXA-48}$-like, and $bla_{GES}$, respectively were found according to the standard BLAST. Moreover, all data on the geographical regions, isolation sources, years and hosts of harboring crabapenemases of isolates have been shown in Table 1 and S1 File.

The size of the complete plasmids ranged from 726 bp to 583,215 bp. The most prevalent alleles found in carbapenemase genes were $bla_{KPC-2}$ (997/1494), $bla_{NDM-1}$ (415/627), $bla_{OXA-48}$ (316/741), and $bla_{GES-5}$ (11/21). All allele types are shown in Fig 2 and S2 File. One thousand one hundred forty (1140/2254, 50.5%) plasmids were potentially conjugative and carried all four conjugal components including *oriT*, relaxase, type IV coupling protein (T4CP), and type IV secretion system (T4SS). The $bla_{KPC}$ was mostly located on the plasmids with IncFII/IncR replicon types (332/1132, 29.4%). The $bla_{NDM}$ gene was mainly related to the plasmids with the IncX3 replicon type (111/495, 22.6%). While, 434/617, 70.3% of plasmids with $bla_{OXA-48}$ had the IncL replicon type.

### 3.2. The co-existence of other antimicrobial resistance genes in the plasmids

Different antimicrobial resistance genes against various classes of antibiotics, including extended-spectrum beta-lactams, carbapenems, quinolones, aminoglycosides, chloramphenicol, tetracycline, macrolide, fosfomycin and sulfonamides along with genes encoding efflux pump proteins and antiseptic-resistance genes, were found in plasmids carrying a least one carbapenemase gene. The most prevalent co-existed genes in plasmids harboring $bla_{KPC}$, $bla_{OXA}$, $bla_{NDM}$, and $bla_{GES}$ were $bla_{TEM-1}$, $bla_{OXA-232}$, $ble_{MBL}$, and *aac (6′)-Ib4*, respectively. See Fig 3 and S3 File.

### 3.3. The co-occurrence of carbapenemase genes

Analysis of the data retrieved from the GenBank database revealed that the co-occurrence of carbapenemase genes in different plasmids but in the same strains. This coincidence was found in forty-two genomes. The co-occurrence of carbapenemase genes with predominant allele types was as follows. A number of seventeen plasmids had $bla_{NDM-1}$ and $bla_{KPC-2}$. Four plasmids had $bla_{OXA-48}$/$bla_{NDM-1}$, and three plasmids had $bla_{OXA-48}$/$bla_{KPC-2}$. The rest of the plasmids having the co-occurrence genes has been shown in Table 2. Also four plasmids, including NZ_CP094991, NZ_CP104796.1, NZ_CP086664.1, and NZ_CP090126.1 simultaneously contained three carbapenemase genes, including $bla_{OXA-48}$/$bla_{NDM-1}$/$bla_{KPC-2}$, $bla_{OXA-181}$/$bla_{NDM-1}$/$bla_{NDM-4}$, $bla_{OXA-48}$/$bla_{NDM-1}$/$bla_{KPC-2}$, and $bla_{NDM-1}$/$bla_{KPC-2}$/$bla_{KPC-2}$, respectively. See Table 2. The conjugal plasmids were various among the strains. For example, in NZ_CP050376.1 and NZ_CP041082.1 the plasmids were potentially conjugative, whereas in some other strains, *e.g.* CP065949.1 and NZ_CP024038.1 one plasmid was potentially conjugative and another plasmid was not conjugative. In strains with plasmids containing three

**Table 1. The additional data on the geographical regions, isolation sources, years and hosts of all isolates harboring crabapenemases.**

| Data                          Gene | $bla_{GES}$ | $bla_{OXA-48}$-like | $bla_{NDM}$ | $bla_{KPC}$ |
|---|---|---|---|---|
| **No. of accession numbers** | 9 | 595 | 616 | 1129 |
| **Available Biosample** | 2 | 582 | 410 | 794 |
| **Geographic location** | South Africa (1) | Spain (219)<br>Netherlands (131)<br>Switzerland (42)<br>India (41)<br>China (28)<br>Germany (19)<br>Russia (9)<br>USA (9)<br>Australia (7)<br>Other countries (46)<br>Missing data (31) | China (122)<br>Bangladesh (57)<br>USA (29)<br>Thailand (24)<br>Vietnam (18)<br>India (11)<br>UK (10)<br>South Korea (9)<br>Italy (8)<br>Russia (7)<br>Canada (6)<br>Myanmar (6)<br>Nepal (6)<br>Spain (6)<br>Switzerland (6)<br>Other countries (35)<br>Missing data (50) | China (382)<br>USA (101)<br>Spain (58)<br>Italy (36)<br>Brazil (29)<br>Germany (23)<br>Japan (18)<br>Other countries (60)<br>Missing data (87) |
| **Isolation source** | Trachea (1) | Hospital (201)<br>Urine (33)<br>Blood (32)<br>Rectal swab (17)<br>Sputum (11)<br>Trachea (10)<br>Wound (8)<br>Endotracheal (8)<br>Pus (7)<br>Stool (5)<br>Abdominal (4)<br>Skin (3)<br>Intestine (3)<br>Outdoors (2)<br>Nasal (2)<br>Other (48)<br>Missing data (188) | Blood (57)<br>Urine (37)<br>Wound (36)<br>Sputum (32)<br>Rectal (12)<br>Hospital (8)<br>Feces (8)<br>Trachea (8)<br>River (5)<br>Pus (5)<br>Stool (4)<br>Respiratory (4)<br>Nasal (2)<br>Other (53)<br>Missing data (139) | Sputum (106)<br>Blood (87)<br>Urine (58)<br>Hospital (48)<br>Rectal swab (30)<br>Broncho aspirate (11)<br>Lavage (8)<br>Stool (6)<br>Wound (6)<br>Trachea (5)<br>Abdominal (4)<br>Other (114)<br>Missing data (311) |
| **Collection date** | Missing data (1) | 2018 (276)<br>2019 (90)<br>2017 (49)<br>2014 (40)<br>2015 (22)<br>2016 (20)<br>2020 (17)<br>2013 (15)<br>2022 (6)<br>2021 (4)<br>2012 (4)<br>2011 (1)<br>Massing data (38) | 2017 (85)<br>2016 (53)<br>2019 (44)<br>2018 (34)<br>2015 (30)<br>2013 (22)<br>2014 (20)<br>2021 (18)<br>2012 (14)<br>2020 (11)<br>2022 (6)<br>2011 (3)<br>2010 (3)<br>200 (1)<br>Missing data (66) | 2018 (154)<br>2017 (96)<br>2019 (78)<br>2016 (51)<br>2015 (46)<br>2014 (41)<br>2013 (37)<br>2020 (34)<br>2021 (31)<br>2012 (28)<br>2022 (23)<br>Other years (58)<br>Missing data (117) |

*(Continued)*

**Table 1.** (Continued)

| Data          Gene | *bla*<sub>GES</sub> | *bla*<sub>OXA-48</sub>-like | *bla*<sub>NDM</sub> | *bla*<sub>KPC</sub> |
|---|---|---|---|---|
| **Host** | Homo sapiens (1) | Homo sapiens (521)<br>Dog (7)<br>Cat (2)<br>Flies (2)<br>Canis (1)<br>Missing data (49) | Homo sapiens (314)<br>Canis (2)<br>Chicken (1)<br>Dog (1)<br>Cattel (1)<br>Cat (1)<br>Swine (1)<br>Houseflies (1)<br>Animal (1)<br>Missing data (87) | Homo sapiens (634)<br>Equus (1)<br>Dog (1)<br>Pig (1)<br>Missing data (157) |

carbapenemase genes, the plasmids were potentially conjugative or at least were mobilizable (only lacked *oriT*).

## 3.4. The gene repetition in the plasmids

According to this study, there were 15 plasmids with carbapenemase gene repetitions. See Table 3. In eleven plasmids harboring *bla*<sub>KPC-2</sub> (NC_011383.1, CP107423.1, NZ_OM144977.1, NZ_OL891656.1, NZ_CP066901.1, NZ_CP097691.1, NZ_CP097674.1, and NZ_MT920901.1), *bla*<sub>NDM-1,</sub> (NZ_CP098375.1) and *bla*<sub>GES-24</sub> (LC623933.1, and LC620536.1), the carbapenemase gene had two copy numbers. In addition, in four plasmids, including NZ_MZ512197.1, NZ_CP064771.1, NZ_CP008933.1, and NZ_CP098375.1, three copy numbers of genes had

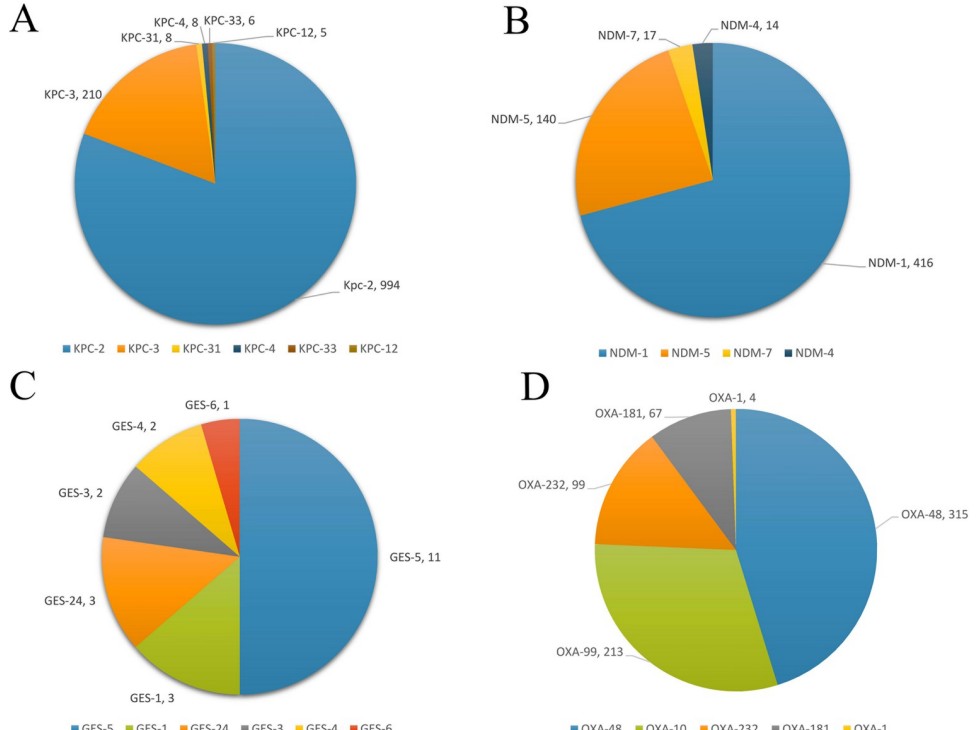

**Fig 2. The prevalence of allele types of the major carbapenemase genes. A)** The different of allele types of *bla*<sub>KPC</sub> gene, **B)** The proportions of all allele types observed in *bla*<sub>NDM</sub> gene, **C)** The frequency of *bla*<sub>OXA-48</sub>-like allele types, **D)** All allele types detected in *bla*<sub>GES</sub> gene.

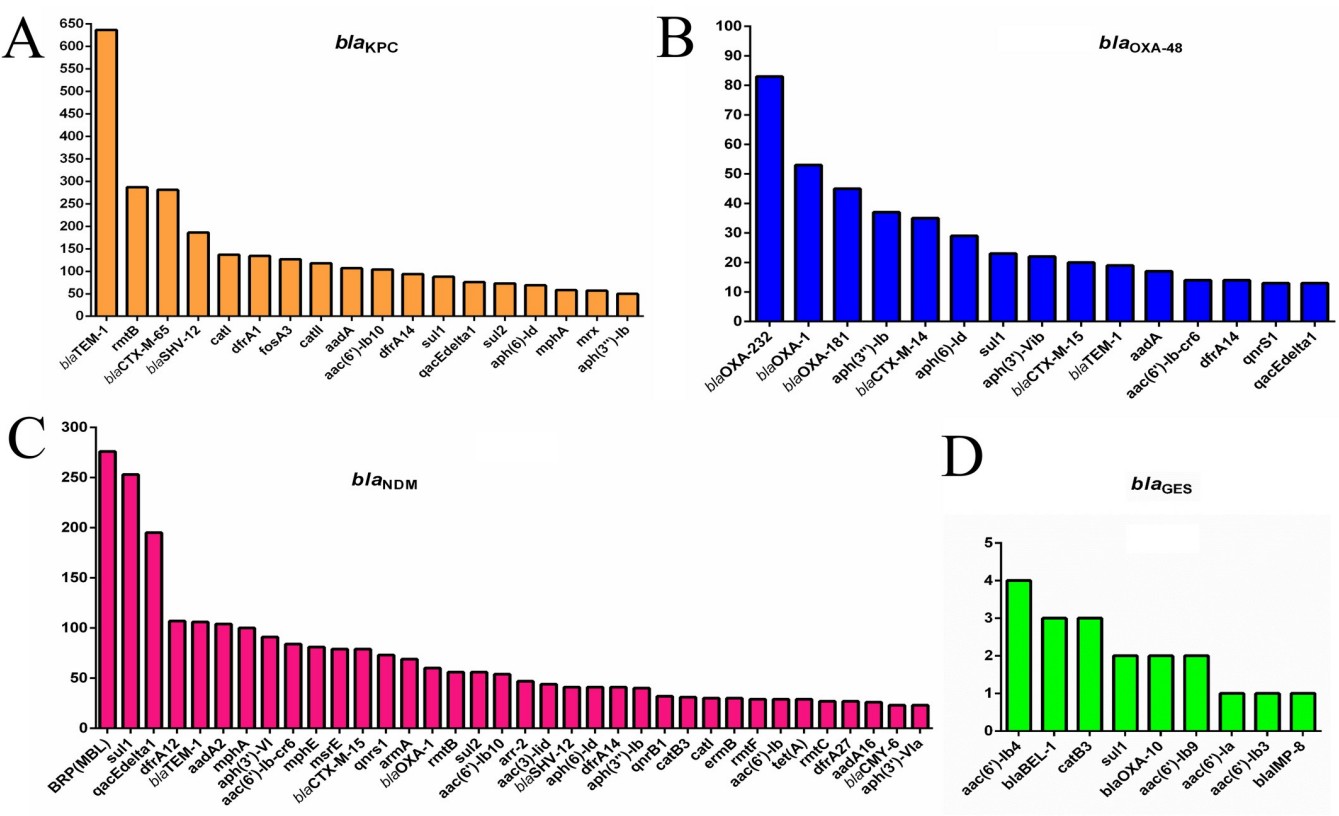

**Fig 3. The prevalence of other antimicrobial resistance in plasmids containing major carbapenemase genes.** The co-existence rate of other antibiotic resistance genes detected in plasmids harboring *bla*KPC gene (**A**), *bla*OXA-48 gene (**B**), *bla*NDM gene (**C**), and *bla*GES gene (**D**).

been detected. No repetition was found in plasmids carrying *bla*OXA-48. The plasmids containing *bla*KPC with two or three copy numbers mostly belonged to ST11. They had IncFII replicon type and was potentially conjugative. While plasmids with *bla*NDM-1 were associated with ST2816 and ST15, had mostly IncFIB, and were potentially conjugative or mobilizable. Both plasmids harboring *bla*GES-24 belonged to ST12, IncFII, and IncCol of plasmids and they didn't have conjugal systems.

### 3.5. The genetic structure of carbapenemase genes

Class 1 integrons, and transposons including Tn3 family transposase, Tn3-like elements, Tn5403 family transposase, Tn7-like elements, and various insertion sequences (IS) were found in plasmids harboring carbapenemase genes. Forty-three 43/2250 (1.9%), 8 (0.25%), and 3 (0.13%) had *intI*1, *intI*2, and *intI*3, respectively. Four hundred and twelve (412/2254, 18.2%) plasmids carried class 1 integrons along with Tn3 family transposase and Tn3-like elements, and 1036/2254 (45.9%) had the other genetic elements, including Tn3 and Tn3-like families.

The *bla*NDM-1 gene was found between IS*30* and Tn3-like transposase genes. Mapping of this transposon revealed that *ble*MBL, *tat*, *cutA*, *groES*, and *groEL* were located adjacent to *bla*NDM-1. The *bla*OXA-48 was located between IS*1*-like and IS*4*-like families and *lysR* was also a neighbor. Mapping of *bla*KPC-2 showed that this carbapenemase gene was flanked by the transposase family IS*Kpn27* and transposase family IS*Kpn6*. Class 1 integron and class 3 integron were found in plasmids containing *bla*GES-5 and *bla*GES-24, respectively. Other genes, including

**Table 2. Genomic data on STs, Carbapenemase genes, other resistance genes and conjugal apparatus among whole genome sequencing of *K. pneumoniae* with co-occurrence of carbapenemase genes.**

| Genome accession number | ST | No. of plasmids | Accession number of plasmids carrying carbapenemases | Carbapenem genes | Other resistant genes | Size of plasmids | Prediction of Inc availability | Conjugation data* | | | Genetic environment |
|---|---|---|---|---|---|---|---|---|---|---|---|
| | | | | | | | | *oriT*/ Relax | T4SS | T4CP | |
| CP065949.1 | 11 | 5 | NZ_CP065954 | $bla_{NDM-1}$ | N/D | 63769 | IncL | 1 | 1 | 1 | IS transposase |
| | | | NZ_CP065952 | $bla_{KPC-2}$ | $bla_{CTX-M-65}$, *cat*II from *Escherichia coli* K-12 | 100684 | IncFII (pHN7A8), IncR | 0 | 0 | 0 | Tn3 family transposase, IS |
| NZ_CP068572 | 11 | 3 | NZ_CP068574 | $bla_{OXA-48}$ | $bla_{CTX-M-14}$, *aph (3")-Ib*, *aph(3')-Vib*, *aph(6)-Id* | 72218 | IncL | 1 | 1 | 1 | Tn3 family transposase,IS transposase |
| | | | NZ_CP068575 | $bla_{KPC-2}$ | $bla_{TEM-1}$ | 86878 | RepB (R1701) | 0/1 | 1 | 1 | IS transposase |
| NZ_CP061957.1 | 11 | 3 | NZ_CP061958 | $bla_{OXA-48}$ | $bla_{CTX-M-14}$, $bla_{TEM-1}$, *aac(3)-Iie*, *aph(3")-Ib*, *aph(3')-Vib*, *aph(6)-Id* | 109135 | IncL, IncR | 1 | 1 | 1 | Tn3 family transposase, Tn3-like element Tn5403 family, IS |
| | | | NZ_CP061960 | $bla_{KPC-2}$ | $bla_{CTX-M-65}$, $bla_{SHV-12}$ | 144349 | IncFII (pHN7A8),IncR | 0 | 0 | 0 | Tn3 family transposase, Tn3-like element TnAs1, IS |
| NZ_CP029689.1 | 11 | 6 | NZ_CP030135 | $bla_{OXA-48}$ | N/D | 65500 | IncL | 1 | 1 | 1 | IS transposase |
| | | | NZ_CP030134 | $bla_{KPC-2}$ | $bla_{CTX-M-65}$ | 60307 | IncR | 0 | 0 | 0 | IS transposase |
| NZ_CP050371.1 | 11 | 4 | NZ_CP050375 | $bla_{OXA-181}$ | N/D | 51140 | ColKP3, IncX3 | 0/1 | 1 | 1 | Tn3 family transposase, Tn3-like element IS3000 family, IS transposase |
| | | | NZ_CP050374 | $bla_{NDM-1}$ | $bla_{CTX-M-15}$, *rmtF*, *aac(6')-Ib*, *arr-2*, *BRP (MBL)*, *aac(3)-IIe* | 193462 | IncFIB (pQil), IncFII (K), RepB (R1701) | 0 | 0 | 0 | Class 1 integron, Tn3 family transposase, Tn3-like element Tn3, IS |
| NZ_CP071279.1 | 14 | 5 | NZ_CP071281 | $bla_{OXA-48}$ | $bla_{CTX-M-14}$, *aph (3")-Ib*, *aph(6)-Id* | 68932 | IncM1 | 1 | 1 | 1 | Tn3 family transposase,IS transposase |
| | | | NZ_CP071280 | $bla_{NDM-1}$ | *aph(3')-VI*, *mphE*, *msrE*, *armA*, *sul1*, *qacEdelta1*, *aadA2*, *dfrA12* | 269326 | IncFIB (pNDM-Mar), IncHI1B (pNDM-MAR) | 0/1 | 1 | 1 | Class 1 integron, Tn3 family transposase, Tn3-like element Tn3, IS |
| NZ_CP097237.1 | 14 | 5 | NZ_CP097241 | $bla_{OXA-232}$ | N/D | 6141 | ColKP3 | 0 | 0 | 0 | N/D |
| | | | NZ_CP097238 | $bla_{NDM-1}$ | *dfrA14*, $bla_{OXA-1}$, *qnrB1*, *aph(3')-VI*, *mphE*, *msrE*, *armA*, *sul1*, *qacEdelta1*, *aadA2*, *dfrA12*, *aac(6')-Ib-cr6* | 283371 | IncFIB (pNDM-Mar), IncHI1B (pNDM-MAR) | 0/1 | 1 | 1 | Class 1 integron, Tn3-like element IS3000 family transposase, Tn3 family transposase, IS |
| NZ_CP078033.1 | 14 | 4 | NZ_CP078037 | $bla_{OXA-232}$ | N/D | 6141 | ColKP3 | 0 | 0 | 0 | N/D |
| | | | NZ_CP078034 | $bla_{NDM-1}$ | $bla_{CTX-M-15}$, $bla_{OXA-9}$, $bla_{TEM-1}$, *sul2*, *dfrA14*, $bla_{OXA-1}$, *qnrB1*, *mphE*, *msrE*, *armA*, *sul1*, *qacEdelta1*, *aadA2*, *dfrA12*, *aac(6')-Ib10*, *aadA*, *aph(3")-Ib*, *aph(6)-Id*, *aac(6')-Ib-cr6* | 319619 | IncFIB (pNDM-Mar), IncHI1B (pNDM-MAR), IncR | 0/1 | 1 | 1 | Class 1 integron, Tn3 family transposase, Tn3-like element Tn3, IS |

*(Continued)*

**Table 2.** (Continued)

| Genome accession number | ST | No. of plasmids | Accession number of plasmids carrying carbapenemases | Carbapenem genes | Other resistant genes | Size of plasmids | Prediction of Inc availability | Conjugation data* | | | Genetic environment |
|---|---|---|---|---|---|---|---|---|---|---|---|
| | | | | | | | | *oriT*/ Relax | T4SS | T4CP | |
| NZ_CP012753.1 | 14 | 2 | NZ_CP012755 | $bla_{OXA-232}$ | N/D | 6141 | ColKP3 | 0 | 0 | 0 | N/D |
| | | | NZ_CP012754 | $bla_{NDM-1}$ | aph(3')-VI, qnrB1, dfrA12, aadA2, qacEdelta1, sul1, armA, msrE, mphE | 273628 | IncFIB (pNDM-Mar), IncHI1B (pNDM-MAR) | 0/1 | 1 | 1 | Class 1 integron, Tn3 family transposase, Tn3-like element Tn3, IS |
| NZ_CP006798.1 | 14 | 4 | NZ_CP006802 | $bla_{OXA-232}$ | N/D | 6141 | ColKP3 | 0 | 0 | 0 | N/D |
| | | | NZ_CP006799 | $bla_{NDM-1}$ | aph(3')-VI, qnrB1, $bla_{OXA-1}$, dfrA14, dfrA12, aadA2, qacEdelta1, sul1, armA, msrE, mphE, AAC(6')-Ib-cr6 | 283371 | IncFIB (pNDM-Mar), IncHI1B (pNDM-MAR) | 0/1 | 1 | 1 | Class 1 integron, Tn3 family transposase, Tn3-like element IS3000, IS |
| NZ_CP050376.1 | 15 | 5 | NZ_CP050381 | $bla_{OXA-244}$ | N/D | 71402 | IncFII (pCoo) | 1 | 1 | 1 | IS transposase |
| | | | NZ_CP050380 | $bla_{NDM-1}$ | sul2, armA, msrE, mphE, qnrs1, aph(3')-VI, dfrA5, qacEdelta1, mphA, aph(3')-Ia | 353810 | IncFIB (pNDM-Mar), IncHI1B (pNDM-MAR) | 1 | 1 | 1 | Class 1 integron, Tn3 family transposase, Tn3-like element Tn3, IS |
| NZ_CP104796.1 | 16 | 6 | NZ_CP104800 | $bla_{OXA-181}$ | qnrS1 | 51478 | ColKP3, IncX3 | 0/1 | 1 | 1 | IS transposase |
| | | | NZ_CP104795 | $bla_{NDM-1}$ | $ble_{MBL}$, $bla_{SHV-12}$ | 53814 | IncX3 | 0/1 | 1 | 1 | IS transposase |
| | | | NZ_CP104797 | $bla_{NDM-4}$ | $ble_{MBL}$, $ble_{MBL}$ | 193355 | IncFIB (pKPHS1) | 0/1 | 1 | 1 | Tn3-like element Tn5403, IS |
| NZ_CP080362.1 | 16 | 6 | NZ_CP080367 | $bla_{OXA-181}$ | N/D | 51479 | ColKP3, IncX3 | 0/1 | 1 | 1 | Tn3 family transposase, Tn3-like element IS3000 family, IS transposase |
| | | | NZ_CP080366 | $bla_{NDM-4}$ | $ble_{MBL}$, $bla_{TEM-1}$, rmtB | 85191 | IncFII (Yp) | 0/1 | 1 | 1 | Tn3-like element Tn3, IS |
| NZ_CP058940.1 | 16 | 7 | NZ_CP058945 | $bla_{OXA-181}$ | N/D | 50126 | ColKP3, IncX3 | 0/1 | 1 | 1 | Tn3 family transposase, Tn3-like element IS3000 family, IS transposase |
| | | | NZ_CP058942 | $bla_{NDM-5}$ | $bla_{TEM-1}$, rmtB, sul1, dfrA12, ErmB, mphA, qacJ, aadA2 | 99664 | IncFII | 1 | 1 | 1 | Class 1 integron, Tn3 family transposase, Tn3-like element TnAs1, IS |
| NZ_CP041927.1 | 16 | 4 | NZ_CP041931 | $bla_{OXA-181}$ | N/D | 51479 | ColKP3, IncX3 | 0/1 | 1 | 1 | Tn3 family transposase, Tn3-like element IS3000 family, IS transposase |
| | | | NZ_CP041930 | $bla_{NDM-4}$ | $ble_{MB}$, $bla_{TEM-1}$, rmtB | 86019 | IncFII (Yp) | 0/1 | 1 | 1 | Tn3 family transposase, Tn3-like element Tn5403, IS |

(*Continued*)

**Table 2.** (*Continued*)

| Genome accession number | ST | No. of plasmids | Accession number of plasmids carrying carbapenemases | Carbapenem genes | Other resistant genes | Size of plasmids | Prediction of Inc availability | Conjugation data* | | | Genetic environment |
|---|---|---|---|---|---|---|---|---|---|---|---|
| | | | | | | | | *oriT*/Relax | T4SS | T4CP | |
| NZ_CP024038.1 | 16 | 6 | NZ_CP024042 | $bla_{OXA-232}$ | N/D | 6141 | ColKP3 | 0 | 0 | 0 | N/D |
| | | | NZ_CP024039 | $bla_{NDM-1}$ | *dfrA12, aadA2, qacEdelta1, sul1, tet(B), tetR* | 125285 | IncFIA | 1 | 1 | 1 | Class 1 integron, Tn3 family transposase, Tn3-like element IS3000, IS |
| NZ_CP086447.1 | 101 | 7 | NZ_CP086451 | $bla_{OXA-48}$ | N/D | 63589 | IncL | 1 | 1 | 1 | IS transposase |
| | | | NZ_CP086448 | $bla_{NDM-1}$ | *sul2, cmy-16, sul1, qnrA6, qacEdelta1, aadA2, dfrA12, mphE, msrE, armA, aph(3')-VI, $bla_{OXA-10}$, cmlA5, arr-2, dfrA14, floR, tet(A), aph(6)-Id, aph(3'')-Ib* | 189866 | IncC | 1 | 1 | 1 | Class 1 integron, Tn3 family transposase, IS |
| NZ_CP050360.1 | 147 | 10 | NZ_CP050368 | $bla_{OXA-181}$ | N/D | 6812 | ColKP3 | 0 | 0 | 0 | Tn3-like element Tn5403 family |
| | | | NZ_CP050367 | $bla_{NDM-5}$ | *dfrA12, qacEdelta1,sul1, rmtB,TEM-1, aadA2,mphA, ermB* | 103085 | IncFII | 1/1 | 1 | 1 | Tn3-like element TnAs1 family transposase,Tn3 family transposase,class 1 integron, IS |
| NZ_CP077779.1 | 377 | 5 | NZ_CP077782 | $bla_{OXA-48}$ | N/D | 63589 | IncL | 1 | 1 | 1 | IS transposase |
| | | | NZ_CP077784 | $bla_{NDM-1}$ | *sul1, qacEdelta1, dfrA5, aph(3')-VI, qnrs1, mphE, msrE, armA, sul2, mphA* | 348342 | IncFIB (pNDM-Mar), IncHI1B (pNDM-MAR) | 1 | 1 | 1 | Class 1 integron, Tn3 family transposase, Tn3-like element TnAs1, IS |
| NZ_CP091813.1 | 383 | 5 | NZ_CP091815 | $bla_{OXA-48}$ | $bla_{CTX-M-14}$*, aph(3'')-Ib, aph(3')-Vib, aph(6)-Id* | 72224 | IncL | 1/1 | 1 | 1 | Tn3 family transposase, IS |
| | | | NZ_CP091814 | $bla_{NDM-5}$ | *aph(3')-VI, qnrS1, $bla_{CTX-M-15}$, $bla_{TEM-1}$, sul2, armA, msrE, mphE, dfrA5, qacEdelta1, sul1, mphA, aac(6')-Ib10, aadA, aph(3')-Ia* | 376754 | IncFIB (pNDM-Mar), IncHI1B (pNDM-MAR) | 1 | 1 | 1 | Class 1 integron, Tn3 family transposase, Tn3-like element TnAs1, IS |
| CP034200.2 | 383 | 2 | CP034202 | $bla_{OXA-48}$ | $bla_{CTX-M-14}$*, aph(6)-Id, aph(3')-VIb, aph(3'')-Ib* | 72057 | IncL | 1 | 1 | 1 | Tn3 family transposase, IS transposase |
| | | | CP034201 | $bla_{NDM-5}$ | *qnrS1, $bla_{CTX-M-15}$, $bla_{TEM-1}$, sul1, qacEdelta1, dfrA5, mphE, msrE, armA, sul2, aph(3')-Via, aac(6')-Ib10, aadA, aph(3')-Ia* | 372826 | IncFIB (pNDM-Mar), IncHI1B (pNDM-MAR) | 1 | 1 | 1 | Tn3-like element TnAs3, Tn3-like element Tn3, IS |

(*Continued*)

**Table 2.** (Continued)

| Genome accession number | ST | No. of plasmids | Accession number of plasmids carrying carbapenemases | Carbapenem genes | Other resistant genes | Size of plasmids | Prediction of Inc availability | Conjugation data* | | | Genetic environment |
|---|---|---|---|---|---|---|---|---|---|---|---|
| | | | | | | | | *oriT*/Relax | T4SS | T4CP | |
| NZ_CP094991 | 39 | 5 | NZ_CP094992 | $bla_{OXA-48}$ | *catI, $bla_{OXA-1}$, $bla_{TEM-1}$, qnrS1, $bla_{CTX-M-15}$, ant (2")-Ia, qacEdelta1, sul1, aac(6')-Ib-cr6, aadA* | 326663 | IncHI1B (pNDM-MAR) | 1 | 1 | 1 | Class 1 integron integrase, IS transposase |
| | | | NZ_CP094993 | $bla_{NDM-5}$ | *arr-3, cmlA5, $bla_{OXA-10}$, qacEdelta1, sul1, armA, msrE, dfrA12, aadA2, $ble_{MBL}$, $bla_{OXA-1}$, ant(3")-IIa, mphE, aac(6')-Ib-cr6, aac(3)-IIe* | 174840 | IncC | 1 | 1 | 1 | Class 1 integron, Tn3 family transposase, IS |
| | | | NZ_CP094994 | $bla_{KPC-2}$ | N/A | 102252 | IncFIB (pQil), IncFII (K) | 1 | 1 | 1 | Tn3-like element Tn4401, IS |
| NZ_CP086664.1 | 39 | 6 | NZ_CP086665 | $bla_{OXA-48}$ | *catI, $bla_{OXA-1}$, $bla_{TEM-1}$, qnrS1, $bla_{CTX-M-15}$, ant (2")-Ia, qacEdelta1, sul1, aac(6')-Ib-cr6, aadA* | 323074 | IncHI1B (pNDM-MAR) | 1 | 1 | 1 | Class 1 integron integrase IntI1, Tn3 family transposase,IS transposase |
| | | | NZ_CP086668 | $bla_{KPC-2}$ | N/A | 102252 | IncFIB (K), IncFIB (pQil), IncFII(K) | 1 | 1 | 1 | Tn3-like element Tn4401, IS |
| | | | NZ_CP086666 | $bla_{NDM-1}$ | *arr-3, cmlA5, $bla_{OXA-10}$, qacEdelta1, sul1, armA, msrE, dfrA12, aadA2, $ble_{MBL}$, $bla_{OXA-1}$, ant(3")-IIa, mphE, aac(6')-Ib-cr6, aac(3)-IIe* | 174841 | IncC | 1 | 1 | 1 | Class 1 integron, Tn3 family transposase, IS |
| NZ_CP041082.1 | 101 | 7 | NZ_CP041085 | $bla_{OXA-48}$ | N/D | 63499 | IncL | 1 | 1 | 1 | IS transposase |
| | | | NZ_CP041083 | $bla_{NDM-1}$ | *sul2, cmy-16, $bla_{CTX-M-15}$, mphE, msrE, qacEdelta1, $bla_{OXA-10}$, cmlA5, arr-2, floR, tet (A), aph(6)-Id, aph(3")-Ib, armA, aph(3')-VI* | 179254 | IncC | 1 | 1 | 1 | Class 1 integron, Tn3 family transposase, IS |
| NZ_CP101776.1 | 11 | 7 | NZ_CP101780 | $bla_{NDM-1}$ | $ble_{MBL}$, $bla_{SHV-12}$ | 53988 | IncX3 | 0/1 | 1 | 1 | Tn3, IS |
| | | | NZ_CP101782 | $bla_{KPC-2}$ | $bla_{CTX-M-65}$, $bla_{TEM-1}$, rmtB, $bla_{SHV-12}$ | 111949 | IncFII (pHN7A8), | 1/0 | 1 | 0 | Tn3, IS |

(*Continued*)

**Table 2.** (Continued)

| Genome accession number | ST | No. of plasmids | Accession number of plasmids carrying carbapenemases | Carbapenem genes | Other resistant genes | Size of plasmids | Prediction of Inc availability | Conjugation data* | | | Genetic environment |
|---|---|---|---|---|---|---|---|---|---|---|---|
| | | | | | | | | *oriT*/ Relax | T4SS | T4CP | |
| NZ_CP090203.1 | 11 | 3 | NZ_MZ546616 | $bla_{NDM-1}$ | cmy-6, arr-3, dfrA27, qacEdelta1, sul1, rmtC, $ble_{MBL}$, aadA16 | 140306 | IncC | 1 | 1 | 1 | Class 1 integron, Tn3 family transposase, IS |
| | | | NZ_MZ546615 | $bla_{KPC-2}$ | $bla_{CTX-M-65}$, $bla_{TEM-1}$, rmtB, $bla_{SHV-12}$ | 126203 | IncFII (pHN7A8) | 1/0 | 1 | 0 | Tn3,IS |
| NZ_CP092656.1 | 11 | 3 | NZ_CP092653 | $bla_{NDM-1}$ | mphE, msrE, armA, sul1, qacEdelta1, aadA5, dfrA1, $ble_{MBL}$, fosA3, $bla_{SHV-12}$, qnrB4, dha-1, qnrB2, mphA | 355489 | N/A | 0/1 | 1 | 1 | Class 1 integron, Tn3 family transposase, Tn3-like element TnAs1, IS |
| | | | NZ_CP092655 | $bla_{KPC-2}$ | dfrA14, arr-3, qnrS1, $bla_{TEM-1}$, $bla_{CTX-M-3}$, aac (6')-Ib10 | 71683 | IncU | 1 | 1 | 1 | Class 1 integron, Tn3 family transposase, Tn3 like, IS |
| NZ_CP091846.1 | 11 | 4 | NZ_CP091847 | $bla_{NDM-1}$ | mphE, msrE, armA, sul1, qacEdelta1, aadA5, dfrA1, $ble_{MBL}$, fosA3, $bla_{SHV-12}$, qnrB4, dha-1, qnrB2, mphA | 349120 | N/A | 0/1 | 1 | 1 | Class 1 integron, Tn3 family transposase, Tn3-like element TnAs1, IS |
| | | | NZ_CP091849 | $bla_{KPC-2}$ | dfrA14, arr-3, qnrS1, $bla_{TEM-1}$, $bla_{CTX-M-3}$, aac (6')-Ib10 | 72015 | IncU | 1 | 1 | 1 | Class 1 integron, Tn3 family transposase, Tn3 like, IS |
| NZ_CP039819.1 | 11 | 3 | NZ_CP039821 | $bla_{NDM-1}$ | dfrA14, $ble_{MBL}$ | 63046 | IncN | 1 | 1 | 1 | Class 1 integron, IS |
| | | | NZ_CP039820 | $bla_{KPC-2}$ | $bla_{CTX-M-65}$, fosA3, $bla_{TEM-1}$, rmtB, catII from E. coli K-12 | 172001 | IncFII (pHN7A8), IncR | 1 | 1 | 1 | Tn3 family transposase, IS |
| NZ_CP039808.1 | 11 | 4 | NZ_CP039811 | $bla_{NDM-1}$ | $ble_{MBL}$, $bla_{SHV-12}$ | 53097 | IncX3 | 0/1 | 1 | 1 | Tn3 family transposase, Tn3-like element IS3000, IS |
| | | | NZ_CP039810 | $bla_{KPC-2}$ | $bla_{CTX-M-65}$, $bla_{TEM-1}$, rmtB | 153556 | IncFII (pHN7A8), IncR | 1 | 1 | 1 | Tn3-like element TnAs1 family, IS |
| CP034327.1 | 11 | 4 | CP034323 | $bla_{NDM-1}$ | $ble_{MBL}$ | 53144 | IncX3 | 0/1 | 1 | 1 | Tn3 family transposase, Tn3-like element IS3000, IS |
| | | | CP034324 | $bla_{KPC-2}$ | $bla_{SHV-12}$, rmtB, $bla_{TEM-1}$, FosA3, $bla_{CTX-M-65}$, catII from E. coli K-12 | 159467 | IncFII (pHN7A8), IncR | 1 | 1 | 1 | Tn3 family transposase, IS transposase |

*(Continued)*

**Table 2.** (Continued)

| Genome accession number | ST | No. of plasmids | Accession number of plasmids carrying carbapenemases | Carbapenem genes | Other resistant genes | Size of plasmids | Prediction of Inc availability | *oriT*/ Relax | T4SS | T4CP | Genetic environment |
|---|---|---|---|---|---|---|---|---|---|---|---|
| | | | | | | | | Conjugation data* | | | |
| CP034327.1 | 11 | 4 | CP034323 | $bla_{NDM-1}$ | $ble_{MBL}$ | 53144 | IncX3 | 0/1 | 1 | 1 | Tn3 family transposase, Tn3-like element IS3000, IS |
| | | | CP034324 | $bla_{KPC-2}$ | $bla_{SHV-12}$, $rmtB$, $bla_{TEM-1}$, $fosA3$, $bla_{CTX-M-65}$, $cat$II from *E. coli* K-12 | 159467 | IncFII (pHN7A8), IncR | 1 | 1 | 1 | Tn3 family transposase, IS transposase |
| CP109983.1 | 15 | 8 | NZ_CP109986 | $bla_{NDM-1}$ | $ble_{MBL}$ | 86272 | IncFII (Yp) | 1/1 | 1 | 1 | IS transposase |
| | | | NZ_CP109990 | $bla_{KPC-2}$ | N/D | 7329 | ND | 0/0 | 0 | 0 | IS transposase |
| NZ_CP090126.1 | 15 | 6 | NZ_CP090129 | $bla_{NDM-1}$ | $ble_{MBL}$, $bla_{SHV-12}$ | 53096 | IncX3 | 0/1 | 1 | 1 | Tn3 family transposase, Tn3-like element IS3000, IS |
| | | | NZ_CP090128 | $bla_{KPC-2}$ | $mphE$, $msrE$, $armA$, $sul1$, $qacEdelta1$ | 88164 | IncFII (Yp) | 1 | 1 | 1 | Class 1 integron, Tn3 family transposase, IS |
| | | | NZ_CP090127 | $bla_{KPC-2}$ | N/D | 103807 | IncX6 | 1 | 1 | 1 | Tn3-like element Tn3, IS |
| NZ_CP039813.1 | 15 | 5 | NZ_CP039817 | $bla_{NDM-1}$ | N/D | N/A | 51995 | 1 | 1 | 1 | Tn3-like element Tn5403, IS |
| | | | NZ_CP039815 | $bla_{KPC-2}$ | $bla_{CTX-M-15}$ | 97386 | IncFII (Yp) | 1 | 1 | 1 | Tn3-like element Tn3 family transposase, IS |
| NZ_CP039802.1 | 15 | 5 | NZ_CP039806 | $bla_{NDM-1}$ | N/D | 51995 | IncN2 | 1 | 1 | 1 | Tn3-like element Tn5403, IS |
| | | | NZ_CP039805 | $bla_{KPC-2}$ | $bla_{CTX-M-15}$ | 97386 | IncFII (Yp) | 1 | 1 | 1 | Tn3-like element Tn3 family transposase, IS |
| NZ_CP026586.1 | 86 | 4 | NZ_CP026590 | $bla_{NDM-1}$ | $dfrA14$, $qnrs1$, $ble_{MBL}$ | 49215 | IncN | 0/1 | 1 | 1 | Class 1 integron, IS |
| | | | NZ_CP026589 | $bla_{KPC-2}$ | $bla_{CTX-M-65}$, $bla_{TEM-1}$, $rmtB$, $fosA3$ | 89247 | IncFII (pHN7A8) | 1 | 1 | 0 | Tn3 family transposase, IS |
| NZ_CP091048.1 | 464 | 7 | NZ_CP091052 | $bla_{NDM-1}$ | $bla_{SHV-12}$ | 59349 | IncX3 | 0/1 | 1 | 1 | Tn3 family transposase, Tn3-like element IS3000, IS |
| | | | NZ_CP091049 | $bla_{KPC-2}$ | $aac(3)-IId$, $arr-3$, $dfrA27$, $qacEdelta1$, $sul1$, $qnrB4$, $aadA16$ | 248847 | N/A | 0/1 | 1 | 1 | Class 1 integron, Tn3 family transposase, Tn3-like element TnAs1, IS |
| NZ_CP084986.1 | 2667 | 3 | NZ_CP084987 | $bla_{NDM-1}$ | $arr-3$, $dfrA27$, $qacEdelta1$, $sul1$, $ble_{MBL}$, $mphA$, $aac(3)-IId$, $bla_{TEM-1}$, $aadA16$, $aph(6)-Id$, $aph(3”)-Ib$, $sul2$ | 362034 | IncQ1 | 0/1 | 1 | 1 | Class 1 integron, Tn3 family transposase, IS |
| | | | NZ_CP084988 | $bla_{KPC-2}$ | N/A | 142244 | RepB (R1701) | 1 | 1 | 1 | Tn3-like element ISPa38, Tn3-like element Tn3, Tn3-like element Tn5403, IS |

(*Continued*)

**Table 2.** (Continued)

| Genome accession number | ST | No. of plasmids | Accession number of plasmids carrying carbapenemases | Carbapenem genes | Other resistant genes | Size of plasmids | Prediction of Inc availability | *oriT*/Relax | T4SS | T4CP | Genetic environment |
|---|---|---|---|---|---|---|---|---|---|---|---|
| NZ_CP063147.1 | 2667 | 3 | NZ_CP063149 | $bla_{NDM-1}$ | $bla_{TEM-1}$, *aac(3)-IId*, *mphA*, *sul1*, $ble_{MBL}$, *qacEdelta1*, *qnrB6*, *arr-3*, *sul2*, *aph(3")-Ib*, *aph(6)-Id*, *aadA16*, *dfrA27* | 375474 | IncQ1 | 0/1 | 1 | 1 | Class 1 integron, Tn3 family transposase, IS |
| | | | NZ_CP063148 | $bla_{KPC-2}$ | N/D | 159093 | RepB (R1701) | 1 | 1 | 1 | Tn3-like element Tn5403 family, Tn3-like element Tn3 family, Tn3-like element ISPa38 family, IS |
| NZ_CP039828.1 | 3493 | 3 | NZ_CP039829 | $bla_{NDM-1}$ | *mphE*, *msrE*, *sul1*, *qacEdelta1*, *arr-3*, $bla_{TEM-1}$, $ble_{MBL}$, *catB3*, $bla_{OXA-1}$, *aadA16*, *dfrA27*, *aac(6')-Ib9*, *qacG*, *aac(6')-Ib-cr6*, *catII* from *Escherichia coli* K-12 | 317231 | N/A | 0/1 | 1 | 1 | Class 1 integron, Tn3 family transposase, Tn3-like element TnAs1, IS |
| | | | NZ_CP039831 | $bla_{KPC-2}$ | N/D | 175540 | IncFIA (HI1) | 1 | 1 | 1 | Tn3-like element Tn4401 family transposase, Tn3-like element Tn4401 family resolvase TnpR, Tn3-like element Tn5403 family, Tn3-like element ISPa38 family, IS |
| NZ_CP039823.1 | 3493 | 4 | NZ_CP039824 | $bla_{NDM-1}$ | *mphE*, *msrE*, *sul1*, *qacEdelta1*, *arr-3*, $bla_{TEM-1}$, *catB3*, $bla_{OXA-1}$, *aadA16*, *dfrA27*, *aac(6')-Ib9*, *qacG*, *aac(6')-Ib-cr6*, *catII* from *E. coli* K-12 | 318848 | N/D | 0/1 | 1 | 1 | Class 1 integron, Tn3 family transposase, Tn3-like element Tn3, IS |
| | | | NZ_CP039826 | $bla_{KPC-2}$ | N/D | 175540 | IncFIA (HI1) | 1 | 1 | 1 | Tn3-like element Tn4401 family transposase, Tn3-like element Tn4401 family resolvase TnpR, Tn3-like element Tn5403 family, Tn3-like element ISPa38 family, IS |

*1 means presence and 0 means absence

**Table 3. Gene repetition of carbapenemases which has been found from the plasmids.**

| Gene | Accession number | Repetition |
|---|---|---|
| $bla_{KPC}$ | NC_011383.1 | 2 |
| | CP107423.1 | 2 |
| | NZ_OM144977.1 | 2 |
| | NZ_OL891656.1 | 2 |
| | NZ_CP066901.1 | 2* |
| | NZ_MZ512197.1 | 3* |
| | NZ_CP097691.1 | 2 |
| | NZ_CP097674.1 | 2 |
| | NZ_CP064771.1 | 3 |
| | NZ_MT920901.1 | 2* |
| $bla_{NDM}$ | CP030858.1 | 3 |
| | NZ_CP008933.1 | 3 |
| | NZ_CP098375.1 | 2 |
| $bla_{GES}$ | LC623933.1 | 2 |
| | LC620536.1 | 2 |

*Gene duplication was detected in DNA sequence with point mutations and frameshifts.

*aac (6)-Ia*, *cat*, *ant*, *DUF86*, *invA*, and *blaA* were found adjacent to $bla_{GES-24}$ and *dfrB1*, $bla_{OXA-10,}$ *aac(6')-Ib4* and *qacE* delta1 were seen near to $bla_{GES-5}$. The highly prevalent genetic structures associated with the carbapenemase genes have been shown in Fig 4.

## 3.6. Plasmid analysis

The sequence comparisons showed that plasmids carrying $bla_{KPC}$ and $bla_{OXA-48}$ appear to be more homogeneous and conserved, whereas the plasmids carrying $bla_{NDM}$ were more heterogenic and distributed in the circular dendrogram. See Fig 5.

The plasmids harboring $bla_{KPC}$ which were located in the same cluster mostly had IncFII replicon types and were mainly potentially conjugative. In addition, the plasmids with $bla_{OXA-232}$ had the ColKP3 replicon type and were all non-conjugative, whereas $bla_{OXA-48}$ had the IncL replicon type and were all potentially conjugative. Among plasmids harboring $bla_{NDM}$, the conjugation pattern and Inc type were different. Plasmids in the same cluster had the same replicon type (e.g., IncC) but a different Inc type in comparison to plasmids that were in a different cluster (which had IncFIB replicon type). It should also be noted that some of these plasmids that were in one cluster, were all mobilized. While the others in the same cluster had a different conjugal pattern.

## 3.7. Clonal relatedness of strains harboring carbapenemase genes

According to the data, most of the $bla_{KPC}$ (270/1132, 23.8%) and $bla_{NDM}$ (33/495, 6.6%) genes were located on plasmids belonging to ST11. The $bla_{OXA-48}$ (64/617, 10.3%) was almost exclusively carried on plasmids belonging to ST14, and $bla_{GES}$ (3/10, 30%) was associated with ST12. Some STs were associated with only one carbapenemase gene. For example, ST258 was associated with $bla_{KPC,}$ ST1 with $bla_{NDM,}$ and ST231 associated with $bla_{OXA-48}$-like. On the other hand, some other STs, including ST37, ST35, ST16, ST392, ST147, ST17, ST23, ST101, ST307, ST11, ST14, and ST437 were multi-harboring carbapenemase genes and had at least one carbapenemase gene, including $bla_{KPC,}$ $bla_{NDM,}$ and $bla_{OXA-48}$. See Fig 6.

**A**

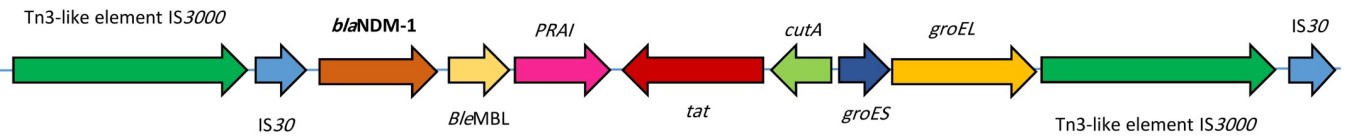

**B**

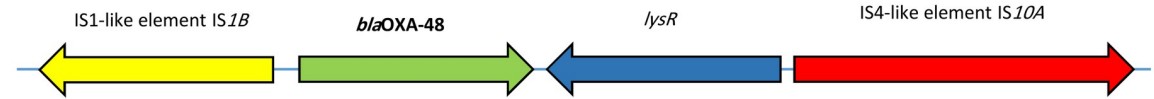

**C**

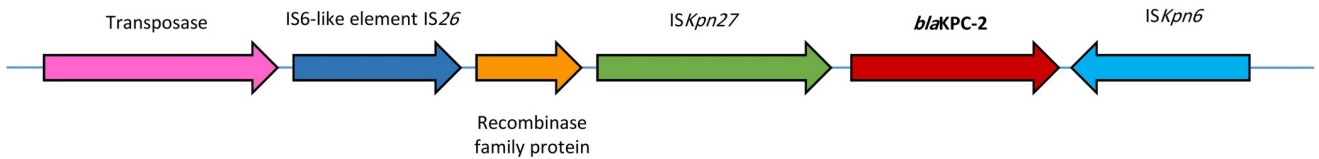

**D**

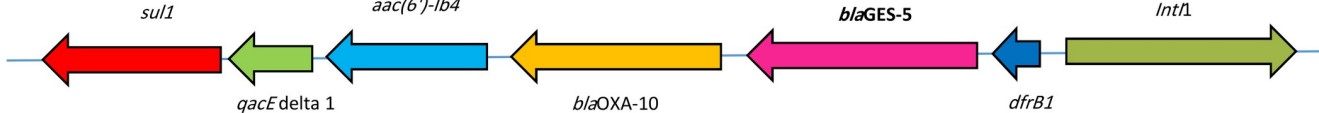

**E**

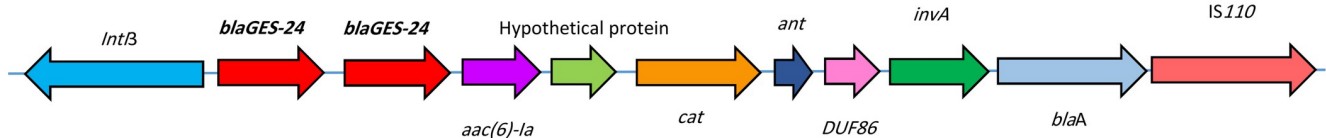

**Fig 4. The genetic environments of major carbapenemase genes associated with class 1 integrons, insertion sequences and transposons. A)** The genetic features of $bla_{NDM-1}$ flanked by Tn3 like elements. The $bla_{NDM-1}$ comes together by other resistance genes, including *tat* and $ble_{MBL,}$ localized on the Tn3-like elements. **B)** The genetic environment of $bla_{OXA-48}$ is between IS*1*-like and IS*4*-like families and *lysR* is also a neighbor. **C)** The $bla_{KPC-2}$ flanks between IS*Kpn6* and IS*Kpn27*. The core structure of IS*Kpn27*/IS*Kpn7-dnaA-*$bla_{KPC-2}$-IS*Kpn6* is highly epidemic in KPC-producing *K. pneumoniae* isolates. **D)** The $bla_{GES-5}$ associated with *intI*1, **E)** While genetic features of $bla_{GES-24}$ related to *intI*3.

Plasmids containing $bla_{KPC}$ that belonged to ST11 were mainly $bla_{KPC-2}$ (258/270, 95.5). However, $bla_{KPC-33}$ (2/270, 0.7%), $bla_{KPC-12}$ (4/270, 1.48%), $bla_{KPC-3}$ (2/270, 0.7%), $bla_{KPC-71}$ (1/270, 0.3%), $bla_{KPC-74}$ (1/270, 0.3%), and $bla_{KPC-93}$ (2/270, 0.7%) were also found sporadic.

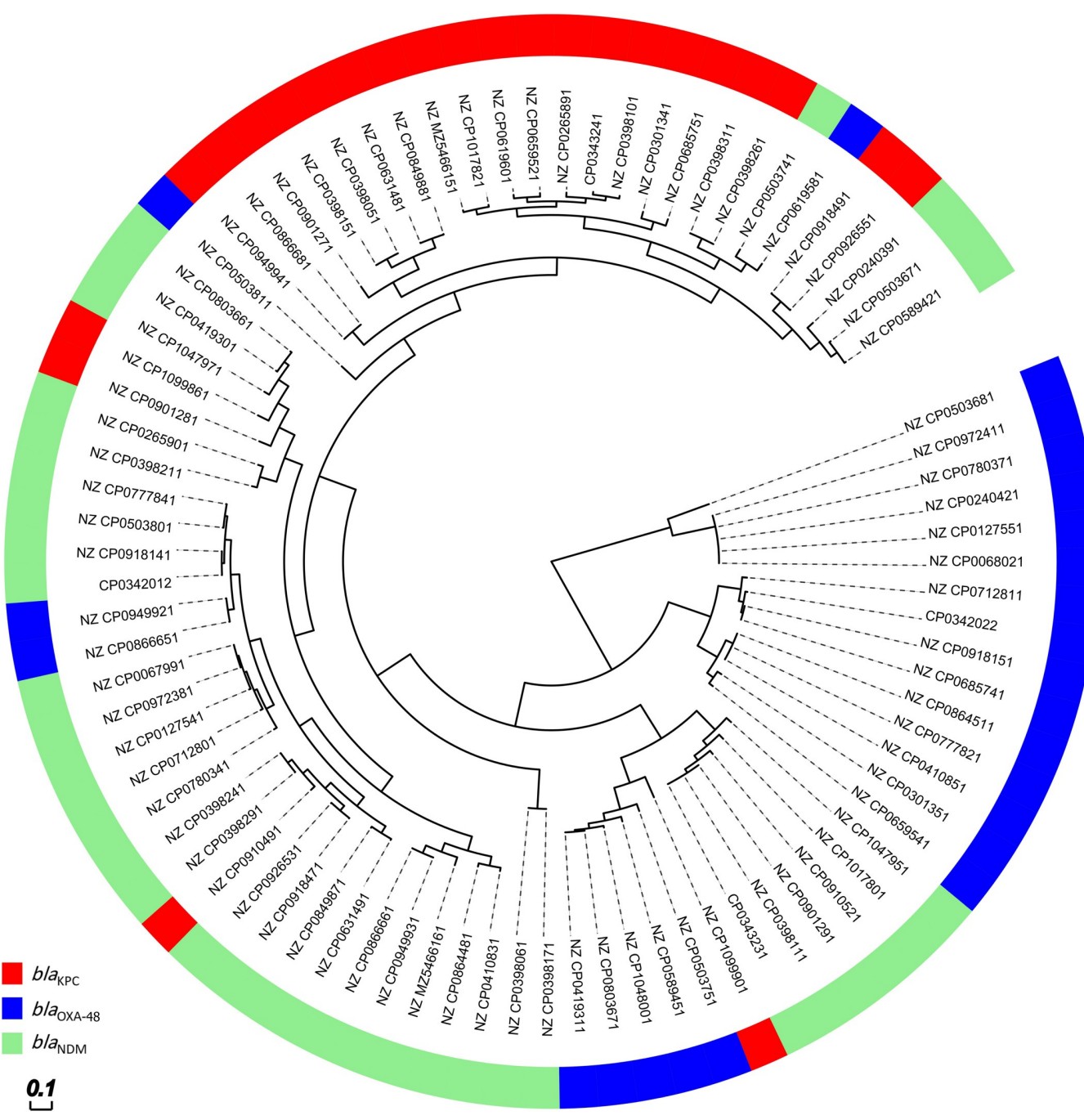

**Fig 5. The circular sequence alignment of eighty-five plasmids harboring multiple major carbapenemase genes.** Data shows that plasmids carrying $bla_{\text{KPC}}$ are more convergent. On the hand, $bla_{\text{NDM}}$ encoding plasmids are divergent and have different genetic characteristics (molecular weight, replicon typing, conjugal apparatus and other antimicrobial genes).

These plasmids were mainly on IncFII and IncR replicon types and 107/270 (39.6%) of plasmids were potentially conjugative. Plasmids with $bla_{\text{NDM}}$ belonging to ST11 were mainly $bla_{\text{NDM-1}}$ (24/33, 72.7%), however, $bla_{\text{NDM-5}}$ (9/33, 27.2%) was also found. These plasmids were mainly on IncX3 and IncC replicon types and 18/33 (54.5%) of plasmids were potentially conjugative while 13/33 (39.3%) were mobilized. Plasmids with $bla_{\text{OXA-48}}$-like belonging to

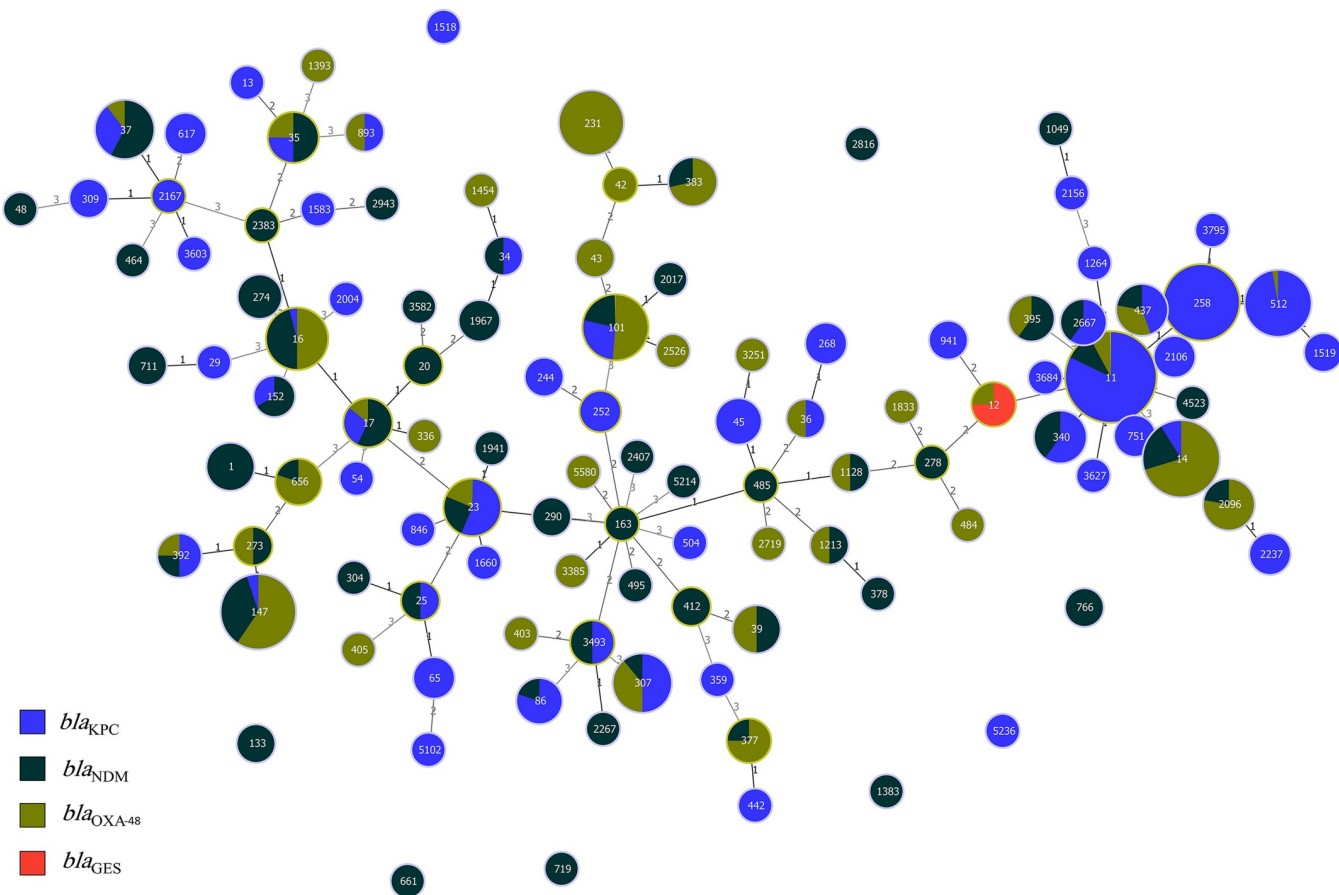

**Fig 6. The minimum spanning tree (MST) of STs carrying the plasmids containing major carbapenemase genes (*e.g. bla*KPC, *bla*NDM, *bla*OXA-48 and *bla*GES) with similarity cut off ≥4 allelic types based on multi-locus sequence typing (MLST) scheme.** ST11, ST14, ST437, ST23, ST307, ST101, ST147, ST16, ST17, ST35, and ST37 are multi-harboring carbapenemases STs in different strains.

ST14 were mainly *bla*OXA-10 (46/64, 71.8%), however, *bla*OXA-232 (14/64, 21.8%), *bla*OXA-48 (2/64, 0.03%), and *bla*OXA-9 (2/64, 0.03%) were also found. It should be noted that in all 46 plasmids with *bla*OXA-10 had a co-existence with *bla*OXA-48. These plasmids were mainly on InL replicon types and 45/64 (70.3%) of plasmids were potentially conjugative. Plasmids with *bla*GES belonging to ST12 were *bla*GES-24. These plasmids had IncCol and IncFII replicon types. None of these plasmids were potentially conjugative.

## 4. Discussion

Carbapenem-resistant *K. pneumoniae* is one of the most challengeable causes of community-acquired and nosocomial infections which can increase morbidity and mortality rate [21]. Multidrug-resistant *K. pneumoniae* isolates complicate treatment, and carbapenems are one of the last-line agents to combat these infections. Therefore, carbapenem resistance could make the situation worse [22]. The presence of carbapenemase genes on conjugative plasmids also contributes to the higher dissemination rates [12]. According to the latest report of CDC and WHO *K. pneumoniae* is considered as urgent priority and the presence of resistance genes, including *bla*KPC2 and *bla*KPC3 along with *bla*NDM and *bla*OXA-48-like as the most prevalent carbapenemase genes could be a worrying issue in public health [23]. In the current study,

bioinformatics tools were used to obtain more information about the genetic characteristics of *K. pneumoniae* plasmids harboring carbapenemase genes.

According to the MST (minimum spanning tree) results, predominated STs in each carbapenemase gene, including $bla_{KPC}$, $bla_{NDM}$, $bla_{OXA}$, and $bla_{GES}$, were ST11, ST14, and ST12. Apart from the predominant STs, some others, including ST37, ST35, ST16, ST392, ST147, ST17, ST23, ST101, ST307, ST11, ST14, and ST437, were multi-harbor sequence types and associated with plasmids containing at least one of the three carbapenemase genes, including $bla_{KPC}$, $bla_{NDM}$, $bla_{OXA-48}$-like.

ST11 is one of the most common ST in *K. pneumoniae* isolates. Recently, outbreaks of ST11 carbapenem-resistant hypervirulent *K. pneumoniae* have been reported in Asian countries, such as China, and ST11 also accounts for 12% of carbapenem-resistant *K. pneumoniae* in Europe. Apart from the wide distribution of ST11 and its isolation from human samples, ST11 could also be isolated from non-human environments, according to Campos-Madueno et al [24]. ST11 is a single-locus variant of ST258, is one of the most common member of CG258, and is more distributed compared to ST258 and ST512 [25–27]. The presence of ESBL encoding genes, including $bla_{CTX-M-65}$ and $bla_{TEM-1}$, which were also seen in this study, is prevalent in ST11 [26]. According to the current study, plasmids harboring $bla_{KPC}$ and $bla_{NDM}$ belonging to ST11. Plasmids with $bla_{KPC}$, especially $bla_{KPC-2}$, mostly had IncFII and IncR replicon types and 39.6% were potentially conjugative. Plasmids with $bla_{NDM}$ had IncX3 and IncC replicon types and 54.5% were potentially conjugative. IncFII is one of the most prevalent plasmids found in carbapenem-resistant isolates and it is restricted to the Enterobacteriaceae family [28]. IncX3 is also found mainly in Enterobacteriaceae, has high transmissibility, and facilitates the spread of $bla_{NDM}$ among *K. pneumonae* [29, 30]. Regarding the current data, the conjugative plasmids harboring carbapenemase and ESBL genes belonging to ST11 could be notable as they could affect the dissemination of resistance genes. On the other hand, due to the genetic structure, $bla_{NDM-1}$ was accompanied by other resistance genes, including *tat* and $ble_{MBL,}$ localized on the Tn3-like elements. The Tn3 family is one of the most important mobile genetic elements with the ability to spread a variety of passenger genes, including those conferring resistance to several classes of antibiotics, including carbapenem and colistin resistance [31]. Mapping of the structure containing $bla_{KPC}$ also revealed IS*Kpn6* and IS*Kpn27* elements. The core structure of IS*Kpn27*/IS*Kpn7-dnaA-bla*$_{KPC-2}$-IS*Kpn6* is highly epidemic in KPC-producing *K. pneumoniae* isolates [32]. The presence of insertion sequences and transposons with high transmission capacity harboring carbapenemase genes in plasmids associated with ST11 *K. pneumoniae* isolates could render these isolates be lethal and cause life-threatening infections.

In the current study, ST14 was another predominant sequence type associated with plasmids containing $bla_{OXA-48.}$ ST14 is one of the major STs carrying multiple resistance genes and is common and associated with pediatric and neonatal infections [33]. Available studies indicate that ST14 *K. pneumoniae* isolates can lead to fatal infections. According to this study, plasmids containing OXA gene (e.g., $bla_{OXA-48}$) and associated with ST14 mostly had IncL replicon type and were conjugative. In addition, mapping of the construct of the cassette containing OXA revealed that this gene is flanked by IS1 and IS4 families. Insertion of these elements into *K. pneumoniae* isolates with notable virulence factors, including *ompK36* results in higher resistance to carbapenems and an increase in the probability of the treatment failures. On the other hand, the adjacency of $bla_{OXA-48}$ with *lysR* can worsen the situation since this gene plays an important role in mediating antibiotic resistance and increasing the virulence of *K. pneumoniae* isolates [34]. In this study, ST12 is another predominant ST associated with plasmids carrying $bla_{GES-24}$ with a notable genetic map. The co-existence of $bla_{GES-24}$ with resistance

genes, including *aac (6)-Ia*, *cat*, and *blaA* genes leads to high resistance to multiple antibiotic classes, maybe increase the rate of treatment failure.

One of the most outstanding findings of the current study is the multi-harbor STs. ST101, ST147, and ST16 are three important STs involved in the spread of *K. pneumoniae* isolates. Overall, the data from the current study showed that the plasmids associated with ST147 and ST101 had mostly plasmid with IncL (with $bla_{OXA}$) and IncFIB and IncFII replicon types (containing $bla_{KPC}$ and $bla_{NDM}$), that were almost potentially conjugative or at least mobilizable. The predominant alleles were $bla_{OXA-48}$, $bla_{KPC-2}$, and $bla_{NDM-1}$, however, $bla_{OXA-10}$, $bla_{OXA-181}$, $bla_{OXA-232}$, $bla_{NDM-5}$, $bla_{NDM-7}$, $bla_{NDM-29}$ and $bla_{NDM-9}$ could also be found in plasmids associated with ST147. These STs are highly associated with lethal infections that according to available studies ST101 could be assumed as a global threat since it plays a major role in the dissemination of resistance genes, including colistin-resistance [35]. ST16 is also a major clone associated with the spread of $bla_{NDM-1}$ and $bla_{OXA232}$ and is thought to be one of the most widespread and high-risk clones worldwide [36]. The results of our study also confirmed these data; however, evaluation of the plasmids examined revealed that ST16 might be associated with plasmids containing $bla_{NDM-4}$ and $bla_{NDM-5}$, $bla_{OXA-48}$, and $bla_{OXA181}$. In addition, according to this study, ST16 was associated with IncFII, IncFIA/B, IncL, and Inc ColKP3. Shukla *et al*, also reported that ST16 carried mainly ColKP3 [37]. Importantly, IncFII, IncFIA/B, and IncL were mostly conjugative or mobilizable. Whereas ColKP3 was mostly non-conjugative. The presence of non-conjugative plasmids in *K. pneumoniae* ST16 may be a noteworthy clue affecting the distribution of carbapenem-resistance genes. Taken together, the presence of these STs with high transmission capacity and multiple carbapenemase genes is a big concern in public health.

Our study also revealed remarkable points regarding the co-existence and co-occurrence of resistance genes. The $bla_{TEM-1}$, $bla_{OXA-232}$, $ble_{MBL}$, and *aac(6')-Ib4* were the most abundant genes in the same plasmids harboring the carbapenem resistance genes. Several genes responsible for resistance to different classes of antibiotics, including aminoglycosides, extended-spectrum beta-lactamase, and carbapenem, were found to coexist with $bla_{KPC}$, $bla_{OXA-48}$, $bla_{NDM}$, and $bla_{GES}$. The co-existence of $ble_{MBL}$ and $bla_{NDM}$ was predictable, as $ble_{MBL}$ is always downstream of $bla_{NDM}$ [38]. The presence of $ble_{MBL}$ responsible for bleomycin resistance (as a glycopeptide antibiotic) with $bla_{NDM}$ could increase the rate of treatment failure. In addition, the presence of two or three plasmids with carbapenem resistance genes among different strains seems to be a critical point. Especially because these plasmids were mainly potentially conjugative and carried transposons such as the Tn3 family and integron class I. Tn3 is one of the most widespread transposase families responsible for the spread of resistance genes [31]. Therefore, these strains with different plasmids, each containing a different resistance gene. Plasmid analysis revealed a high similarity between plasmids containing $bla_{KPC-2}$ and $bla_{OXA-48}$, while $bla_{NDM}$ seems to be heterogeneous. This could be a remarkable point as it shows that $bla_{NDM}$ could be placed in different plasmids with different sequences, which would worsen the situation of antimicrobial resistance.

## 5. Conclusion

The co-existence of different classes of resistance genes, co-occurrence of various carbapenemase genes on separate plasmids, and gene repetition in a plasmid were notable findings of this study. Assessment of genetic characteristics of the plasmids also revealed that $bla_{NDM}$-harboring heterogenic plasmids had a high capacity for dissemination. multi-harbor carbapenemases STs could highly affect the exacerbation of the antimicrobial resistance in this bacterium. *K. pneumoniae* appears to employ multiple genetic strategies for resistance against

carbapenem antibiotics. First, gene repetition and locating carbapenemase genes associated with class 1 and 3 integrons, IS*Kpn* and Tn3 plays important roles in DNA recombination. In addition, the placement of these DNA fragments on transferable plasmids paves the way for widespread expansion of antimicrobial resistance. Finally, the successful and international clones (ST11, ST14, ST437, ST23, ST307, ST101, ST147, ST16, ST17, ST35 and ST37) with high ability to capture carbapenemase-containing plasmids are actually the last circle of this journey which has dramatically increased antibiotic resistance all over the world. It seems that *K. pneumoniae* is collecting various resistance genes and virulence factors with all its genome capacity. Such genetic flexibility of a superbug is not only astonishing but also a very serious health threat. In the future, new methods (*e.g.* vaccination, novel drug targets and antibiotics, and new combination therapy such as antibodies and antibiotics) should be used to fight against *K. pneumoniae*.

## Supporting information

**S1 File. All information of BioSamples from isolates harboring carbapenemases.** (ZIP)

**S2 File. The number of allele types of carbapenemase genes in a plasmid.** (XLSX)

**S3 File. The co-existence of antimicrobial resistance genes in a plasmid.** (XLSX)

**S4 File. The accession numbers of all retrieved plasmids.** (XLSX)

## Acknowledgments

The authors would like to thank the personnel at the Bacteriology Department of the Pasteur Institute of Iran to commence on the manuscript for improvement.

## Author Contributions

**Conceptualization:** Farzad Badmasti.

**Data curation:** Mahshid Khazani Asforooshani, Yeganeh Malek Mohammadi, Mohammad Sholeh, Farzad Badmasti.

**Formal analysis:** Shadi Aghamohammad.

**Methodology:** Shadi Aghamohammad, Mahshid Khazani Asforooshani, Yeganeh Malek Mohammadi.

**Project administration:** Farzad Badmasti.

**Software:** Mohammad Sholeh, Farzad Badmasti.

**Validation:** Farzad Badmasti.

**Visualization:** Farzad Badmasti.

**Writing – original draft:** Shadi Aghamohammad, Mahshid Khazani Asforooshani.

**Writing – review & editing:** Farzad Badmasti.

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
