## [Decision Letter · Decision Letter 0]

31 Jul 2023

PONE-D-23-18985Genetic structure of blaKPC, blaNDM, blaOXA-48, and blaGES genes associated with international clones and conjugative plasmids from Klebsiella pneumoniaePLOS ONE

Dear Dr. Badmasti,

Thank you for submitting your manuscript to PLOS ONE. After careful consideration, we feel that it has merit but does not fully meet PLOS ONE’s publication criteria as it currently stands. Therefore, we invite you to submit a revised version of the manuscript that addresses the points raised during the review process.

We look forward to receiving your revised manuscript.

Kind regards,

Farah Al-Marzooq, MD, PhD

Academic Editor

PLOS ONE

Journal Requirements:

2. Please amend your list of authors on the manuscript to ensure that each author is linked to an affiliation. Authors’ affiliations should reflect the institution where the work was done (if authors moved subsequently, you can also list the new affiliation stating “current affiliation:….” as necessary).

3. We note that Figure 1 in your submission contain copyrighted images. All PLOS content is published under the Creative Commons Attribution License (CC BY 4.0), which means that the manuscript, images, and Supporting Information files will be freely available online, and any third party is permitted to access, download, copy, distribute, and use these materials in any way, even commercially, with proper attribution. For more information, see our copyright guidelines: http://journals.plos.org/plosone/s/licenses-and-copyright.

Additional Editor Comments:

Please, revise the manuscript considering all the reviewers' comments , and try to answer all the queries raised by them

Reviewers' comments:

Reviewer's Responses to Questions

**Comments to the Author**

1. Is the manuscript technically sound, and do the data support the conclusions?

Reviewer #1: Yes

Reviewer #2: Partly

2. Has the statistical analysis been performed appropriately and rigorously? 

Reviewer #1: Yes

Reviewer #2: N/A

3. Have the authors made all data underlying the findings in their manuscript fully available?

Reviewer #1: Yes

Reviewer #2: Yes

4. Is the manuscript presented in an intelligible fashion and written in standard English?

Reviewer #1: Yes

Reviewer #2: No

5. Review Comments to the Author

Reviewer #1: Dear authors

Thank you for your interesting study and well-written/organized paper. I read the manuscript and tried to understand what the study's aims were and how the work was carried out. Using bioinformatics methods, investigators sought to learn more about the genetic characteristics of K.pneumoniae plasmids harboring carbapenemase genes. The research was interesting, and the paper was well-written. However, I have a few minor observations:

• Page 8-15; lines 188-190: Table 1. Genomic data on STs, Carbapenemase genes, other resistance genes, and conjugal apparatus among whole genome sequencing of K.pneumoniae.

COMMENT: It would be greatly appreciated if the authors could indicate the source of /sample type (human specimen, animal specimen, food, environmental, or other sources) for these isolates. This information is crucial for comprehending horizontal gene transfer and its occurrence in humans, animals and other sources.

• Page 22; line 369: “In the future, new methods should be used to fight against K. pneumoniae”.

COMMENT: In order to combat K. pneumoniae in a more effective manner, what novel strategies or approaches do you recommend? Could you perhaps write out in detail your thoughts on this matter?

• Page 22; lines 370-372: “The authors would like to thank the personnel at the Bacteriology Department of the Pasteur Institute of Iran for their help. This research was supported by the Pasteur Institute of Iran.”

COMMENT 1: What kinds of assistance did this staff offer? They should be mentioned explicitly, and they should be acknowledged appropriately.

COMMENT 2: Please provide the sponsor grant number

• Page 22; lines 385-386: “No specific funding was received for this study”.

COMMENT: If you did not receive any funding for this study, why did you state "This research was supported by the Pasteur Institute of Iran"? it must be revised.

• ADDITIONAL COMMENT: Please use K. pneumoniae (italic) instead of K. pneumoniae.

Reviewer #2: This manuscript describes the results of the bioinformatics analysis of the genetic structure of plasmids harbouring carbapenemases in Klebsiella pneumoniae. The plasmids analyzed were completed plasmids and the genes studied were blaKPC, blaNDM, blaOXA-100 and blaGES.

A total of 2254 plasmids harboring carbapenemase genes from GenBank database were analyzed.

The objective of the manuscript is interesting and the results could be useful but the bioinformatics strategy to detect and select the carbapenem resistance genes in plasmids of Klebsiella pneumonia is not sufficiently robust and the results are not studied in deep to extract new findings and significant conclusions.

MAJOR POINTS

- The manuscript is in a very preliminary state.

- The strategy of selection of reference sequences is not very appropriate.

- The method for detecting the carbapenem resistance genes is not correctly designed and probably produces a bias in the detection of genes in the analyzed Klebsiella pneumonia genomes.

In the section “Preparation of initial dataset” of the “Materials and methods” I understand that the authors use BLASTN search for detecting the plasmids harbouring carbapenemase genes. Given that many plasmids have different hosts, a different codon usage, or minimal differences of nucleotide sequence that do not affect the function could jeopardize the detection of functional carbapenemase genes if you only use a strategy based on nucleotide sequence detection. It would be better to use protein sequences of the carbapenemases as query and Genbank sequences of nucleotides as subject database using TBLASTN. TBLASTN searches translated nucleotide databases using a protein query.

Thus, in line Line 287 the authors conclude: “According to the current study, plasmids harboring blaKPC and blaNDM belonging to ST11”. I wonder if the reason of that is that the reference genes used to select the plasmids to be analyzed come from ST11 isolates. That is one of the reason why it is better to do a TBLASTN to search the reference sequences of proteins in nucleotide databases avoiding to have a bias and allowing to select proteins functionally similar but with different nucleotide sequence.

- As a research article the manuscript does not present any important new finding neither some new interpretation of the dataset analyzed. Considered as a review lacks many of the aspects required for a review about carbapenem resistance genes in Klebsiella pneumonia.

- The discussion about the publications related with this manuscript is poor. For example these publications are not included in the references of the manuscript:

Campos-Madueno EI, Moser AI, Jost G, Maffioli C, Bodmer T, Perreten V, Endimiani A. Carbapenemase-producing Klebsiella pneumoniae strains in Switzerland: human and non-human settings may share high-risk clones. J Glob Antimicrob Resist. 2022 Mar;28:206-215.

Karampatakis T, Tsergouli K, Behzadi P. Carbapenem-Resistant Klebsiella pneumoniae: Virulence Factors, Molecular Epidemiology and Latest Updates in Treatment Options. Antibiotics (Basel). 2023 Jan 21;12(2):234. doi: 10.3390/antibiotics12020234. PMID: 36830145; PMCID: PMC9952820.

Karaiskos I, Galani I, Papoutsaki V, Galani L, Giamarellou H. Carbapenemase producing Klebsiella pneumoniae: implication on future therapeutic strategies.Expert Rev Anti Infect Ther. 2022 Jan;20(1):53-69. doi: 10.1080/14787210.2021.1935237. Epub 2021 Jun 3.PMID: 34033499 Review.

- Important analysis as the distribution of the plasmids with carbapenem resistance genes in different hosts and in different human tissues are not analyzed.

- Geographic provenance of the genomes is not included in the analysis

- Some study of the plasmids with carbapenemase genes in close species that share microenvironments is not included. This manuscript is focused on Klebsiella pneumonia but it would be needed to discuss if these carbapenem resistance genes and/or the plasmids that harbour them are also present in other bacterial species sharing host and microenvironment as it could be Escherichia coli, Salmonella or other enterobacteria.

Minor points:

Line 103:

“In addition, we characterized the genetic features of plasmids harboring carbapenemase genes including replicon types, conjugation ability, the coexistence of carbapenem with other antimicrobial resistance genes, co-occurrence of carbapenemase genes in one strain, gene repetition, and phylogenetic relatedness.”

Edit this sentence to clarify and explain in more detail what features of the plasmids harboring carbapenemase genes you had characterized in this work.

Line 107:

“The complete nucleotide sequences of plasmids containing four carbapenemase genes, including blaKPC, blaNDM, blaOXA-48, and blaGES were retrieved from the GenBank database (https://www.ncbi.nlm.nih.gov/genbank/).”

I understand that your criterion of selection of plasmids was to have any of those 4 types of carbapenemase genes (blaKPC, blaNDM, blaOXA-48, and blaGES) not to have all the four types in a plasmid. It that is the case it is not clear in this sentence.

Line 125:

oriTfnder tool (https://bioinfomml.sjtu.edu.cn/oriTf inder/)

Please correct the name of the tool and the url

Line 147:

Indicate which is the plasmid with 5,803,733 bp

Line 164:

“plasmids carrying four mentioned carbapenemase genes.”

This sentence is confusing because I understand that there are not 4 carbapenemase genes in each selected plasmid.

Line 276

“According to the MST results, STs predominated in each carbapenemase gene, including 276 blaKPC, blaNDM, blaOXA, and blaGES, were ST11, ST14, and ST12. Apart from the predominant 277 STs, some others, including ST37, ST35, ST16, ST392, ST147, ST17, ST23, ST101, ST307, 278 ST11, ST14, and ST437, were multi-harbor sequence types and associated with plasmids 279 containing three carbapenemase genes, including blaKPC, blaNDM, blaOXA.”

Please, add (Multilocus Sequence Typing) after MST and rewrite this sentence because its meaning is not clear.

Line 293

K.pneumonae -> K. pneumoniae

In general please put K. pneumoniae in italics (it is not my case but microbiologists suffer a lot if you don’t)

Line 320

“One of the most outstanding findings of the current study is the multi-harbor STs. STs 101, 320 147, and 16 are three important sequence types involved in the spread of K.pneumoniae isolates. Overall, the data from the current study showed that the plasmids associated with ST147 and ST101 had mostly IncL (with blaOXA) and IncFIB and IncFII replicon types (containing blaKPC and blaNDM), that were almost potentially conjugative or at least mobilizable. The predominant alleles were blaOXA-48, blaKPC-2, and blaNDM-1, however, blaOXA-10, blaOXA-181, blaOXA-232, blaNDM-5, blaNDM-7, blaNDM-29 and blaNDM-9 could also be found in plasmids associated with ST147”

This sentence needs to be rewritten.

Line 344

“…plasmids harboring the carbapenem/ genes.” → plasmids harboring the carbapenem resistance genes

Line 361

“The co-existence of resistance genes belonging to different antibiotic clases,….” → The co-existence of different antibiotic classes resistance genes,….

Line 365

“multi-harbor carbapenemases and international STs (e.g. ST11, ST23, ST14, and ST12) could highly affect the exacerbation of the antimicrobial resistance in K. pneumoniae.”

IMHO this sentence has not sense.

Finally, I want to encourage the authors to continue working on it because I think they can do a better job on this topic, which is very important and needs a deep analysis.

6. PLOS authors have the option to publish the peer review history of their article (what does this mean?). If published, this will include your full peer review and any attached files.

Reviewer #1: **Yes: **Dr.Melese Abate Reta (PhD)

Reviewer #2: No

---

## [Author Response · Author response to Decision Letter 0]

24 Aug 2023

View Letter

PONE-D-23-18985

Genetic structure of blaKPC, blaNDM, blaOXA-48, and blaGES genes associated with international clones and conjugative plasmids from Klebsiella pneumoniae

PLOS ONE

Dear Dr. Farah Al-Marzooq,

We would like to thank the reviewers for their precious time to review our submitted manuscript and their helpful and valuable comments. We believe the comments and revisions have made our manuscript a more valuable paper. The comments have been added to the manuscript and the questions have been answered. The changes, as suggested, were track changed in the manuscript. 

Best regards,

Dr. Farzad Badmasti

Journal Requirements:

RE: Thank you so much you have checked it.

2. Please amend your list of authors on the manuscript to ensure that each author is linked to an affiliation. Authors’ affiliations should reflect the institution where the work was done (if authors moved subsequently, you can also list the new affiliation stating “current affiliation:….” as necessary).

RE: It was done. 

3. We note that Figure 1 in your submission contain copyrighted images. All PLOS content is published under the Creative Commons Attribution License (CC BY 4.0), which means that the manuscript, images, and Supporting Information files will be freely available online, and any third party is permitted to access, download, copy, distribute, and use these materials in any way, even commercially, with proper attribution. For more information, see our copyright guidelines: http://journals.plos.org/plosone/s/licenses-and-copyright.

 RE: Thank you for your comment. The Figure has been reproduced. There is no copy-write content in it.

Additional Editor Comments:

Please, revise the manuscript considering all the reviewers' comments , and try to answer all the queries raised by them

Reviewers' comments:

Reviewer's Responses to Questions

Comments to the Author

1. Is the manuscript technically sound, and do the data support the conclusions?

Reviewer #1: Yes

Reviewer #2: Partly

2. Has the statistical analysis been performed appropriately and rigorously?

Reviewer #1: Yes

Reviewer #2: N/A

3. Have the authors made all data underlying the findings in their manuscript fully available?

Reviewer #1: Yes

Reviewer #2: Yes

4. Is the manuscript presented in an intelligible fashion and written in standard English?

Reviewer #1: Yes

Reviewer #2: No

5. Review Comments to the Author

Reviewer #1: Dear authors

Thank you for your interesting study and well-written/organized paper. I read the manuscript and tried to understand what the study's aims were and how the work was carried out. Using bioinformatics methods, investigators sought to learn more about the genetic characteristics of K.pneumoniae plasmids harboring carbapenemase genes. The research was interesting, and the paper was well-written. However, I have a few minor observations:

• Page 8-15; lines 188-190: Table 1. Genomic data on STs, Carbapenemase genes, other resistance genes, and conjugal apparatus among whole genome sequencing of K.pneumoniae.

COMMENT: It would be greatly appreciated if the authors could indicate the source of /sample type (human specimen, animal specimen, food, environmental, or other sources) for these isolates. This information is crucial for comprehending horizontal gene transfer and its occurrence in humans, animals and other sources.

RE: First, we would like to thank you for your very careful review of our paper and for your subsequent comments, corrections, and suggestions as well. The additional data on the source, geographical region, and the year of isolation has been added to the manuscript as Table 1.

• Page 22; line 369: “In the future, new methods should be used to fight against K. pneumoniae”.

COMMENT: In order to combat K. pneumoniae in a more effective manner, what novel strategies or approaches do you recommend? Could you perhaps write out in detail your thoughts on this matter?

RE: It was done.

• Page 22; lines 370-372: “The authors would like to thank the personnel at the Bacteriology Department of the Pasteur Institute of Iran for their help. This research was supported by the Pasteur Institute of Iran.”

COMMENT 1: What kinds of assistance did this staff offer? They should be mentioned explicitly, and they should be acknowledged appropriately.

RE: Thank you for your precise observation. The correction has been done.

COMMENT 2: Please provide the sponsor grant number

• Page 22; lines 385-386: “No specific funding was received for this study”.

COMMENT: If you did not receive any funding for this study, why did you state "This research was supported by the Pasteur Institute of Iran"? it must be revised.

RE: Thank you for your comment. The correction has been done.

• ADDITIONAL COMMENT: Please use K. pneumoniae (italic) instead of K. pneumoniae.

RE: The corrections have been done.

Reviewer #2: This manuscript describes the results of the bioinformatics analysis of the genetic structure of plasmids harbouring carbapenemases in Klebsiella pneumoniae. The plasmids analyzed were completed plasmids and the genes studied were blaKPC, blaNDM, blaOXA-100 and blaGES.

A total of 2254 plasmids harboring carbapenemase genes from GenBank database were analyzed.

The objective of the manuscript is interesting and the results could be useful but the bioinformatics strategy to detect and select the carbapenem resistance genes in plasmids of Klebsiella pneumonia is not sufficiently robust and the results are not studied in deep to extract new findings and significant conclusions.

MAJOR POINTS

- The manuscript is in a very preliminary state.

RE: First, we would like to thank you for your very careful review of our paper. In this study, we attempted to present a correct and accurate genetic map of plasmids containing carbapenem resistance genes. One of our major goals was to accurately represent the distribution of four major carbapenem resistance genes in the predominant sequences types, to show the homogeneity of the plasmid, and to show the diversity of allele types. To the best of our knowledge, the representation of all this information in the genomic layer has been done in fewer articles. Furthermore, the manuscript has been enhanced by incorporating supplementary information pertaining to the origin, geographic location, and year of isolation. We really do believe that this manuscript helps investigators who have been working on cabapenemas in K. pneumoniae to design a project and finding new genetic features.

- The strategy of selection of reference sequences is not very appropriate.

RE: In this study four Refseq accession numbers of major cabapenemases were consider to seek all completed plasmids and DNA fragments using BLASTn. These BLASTs were done based an standard criteria. For example, Microbial BLAST (database: All genomes; Organism: K. pneumoniae; Optimize for: Highly similar sequences; Max target sequences: 5000; Expect threshold: 0.05; Word size: 28; Gap Costs: linear).

- The method for detecting the carbapenem resistance genes is not correctly designed and probably produces a bias in the detection of genes in the analyzed Klebsiella pneumonia genomes.

In the section “Preparation of initial dataset” of the “Materials and methods” I understand that the authors use BLASTN search for detecting the plasmids harbouring carbapenemase genes. Given that many plasmids have different hosts, a different codon usage, or minimal differences of nucleotide sequence that do not affect the function could jeopardize the detection of functional carbapenemase genes if you only use a strategy based on nucleotide sequence detection. It would be better to use protein sequences of the carbapenemases as query and Genbank sequences of nucleotides as subject database using TBLASTN. TBLASTN searches translated nucleotide databases using a protein query.

Thus, in line Line 287 the authors conclude: “According to the current study, plasmids harboring blaKPC and blaNDM belonging to ST11”. I wonder if the reason of that is that the reference genes used to select the plasmids to be analyzed come from ST11 isolates. That is one of the reason why it is better to do a TBLASTN to search the reference sequences of proteins in nucleotide databases avoiding to have a bias and allowing to select proteins functionally similar but with different nucleotide sequence.

RE: As mentioned in the title and also different parts of the manuscript, this study aimed to show the genetic map (DNA layer) of the plasmids harboring carbapenem resistance genes. Finding the dominant sequence carrying resistance genes and trying to determine the degree of heterogeneity of plasmids as well as characterizing a genomic content were the main goals of this study, and all these goals can be realized in the DNA level. For example, the determination of allele type of carabapenemases has been established based on single nucleotide polymorphisms (SNPs) in DNA layer. MLST and ST are based on SNP as well. We think there is no need to be assessed in protein layer. It should also be noted that there seems we were not clear about the sequence types. Actually, ST11 is not the reference ST of our study on which the rest of the data are analyzed, but ST11 was one of the dominant STs in this study.

- As a research article the manuscript does not present any important new finding neither some new interpretation of the dataset analyzed. Considered as a review lacks many of the aspects required for a review about carbapenem resistance genes in Klebsiella pneumonia.

RE: As mentioned above, the main objective of the current study is to reveal the precise genetic map of plasmids harboring carbapenem genes. The accurate drawing of the genetic map of the factors that play a prominent role in the transmission of carbapenem resistance as an important topic that has not yet been fully addressed. In addition, the identification of the predominant STs that may play an important role in the spread of resistance genes, along with the type of alleles they may carry, is an important issue that is prominent in AMR. There are some new findings in this study

1- Predominant allele type of each carbapenemase

2- Evaluation of conjugal apparatus in the plasmids harboring major carbapenemases

3- Detection of co-existence and co-occurrence among the plasmids

4- Determination of all genetic maps, divergence and convergence among the plasmids

5- Detection of gene repetition among carbapenemases

6- Revealing of all STs harboring carbapenemases at a glance

- The discussion about the publications related with this manuscript is poor. For example these publications are not included in the references of the manuscript:

Campos-Madueno EI, Moser AI, Jost G, Maffioli C, Bodmer T, Perreten V, Endimiani A. Carbapenemase-producing Klebsiella pneumoniae strains in Switzerland: human and non-human settings may share high-risk clones. J Glob Antimicrob Resist. 2022 Mar;28:206-215.

Karampatakis T, Tsergouli K, Behzadi P. Carbapenem-Resistant Klebsiella pneumoniae: Virulence Factors, Molecular Epidemiology and Latest Updates in Treatment Options. Antibiotics (Basel). 2023 Jan 21;12(2):234. doi: 10.3390/antibiotics12020234. PMID: 36830145; PMCID: PMC9952820.

Karaiskos I, Galani I, Papoutsaki V, Galani L, Giamarellou H. Carbapenemase producing Klebsiella pneumoniae: implication on future therapeutic strategies.Expert Rev Anti Infect Ther. 2022 Jan;20(1):53-69. doi: 10.1080/14787210.2021.1935237. Epub 2021 Jun 3.PMID: 34033499 Review.

RE: The mentioned references have been added to the manuscript. 

- Important analysis as the distribution of the plasmids with carbapenem resistance genes in different hosts and in different human tissues are not analyzed.

- Geographic provenance of the genomes is not included in the analysis

RE: The additional data on the source, geographical region, and the year of isolation has been added to the manuscript.

- Some study of the plasmids with carbapenemase genes in close species that share microenvironments is not included. This manuscript is focused on Klebsiella pneumonia but it would be needed to discuss if these carbapenem resistance genes and/or the plasmids that harbour them are also present in other bacterial species sharing host and microenvironment as it could be Escherichia coli, Salmonella or other enterobacteria.

RE: The purpose of the present study was to fully and accurately investigate the genetic structure of plasmids carrying the carbapenem resistance gene in Klebsiella pneumoniae as a very important nosocomial pathogen. Considering the high number of plasmids from other genus is complex and requires a very extended and progressed analysis. In fact, the main aim of the current study was to focus on the fully evaluation of genetically aspects of plasmids harboring carbapenemase genes in just one species. The analysis of other bacteria, including Escherichia and Salmonella, and the data collection and comparison is very interesting and appealing. However, it could be done in a different study. 

Minor points:

Line 103:

“In addition, we characterized the genetic features of plasmids harboring carbapenemase genes including replicon types, conjugation ability, the coexistence of carbapenem with other antimicrobial resistance genes, co-occurrence of carbapenemase genes in one strain, gene repetition, and phylogenetic relatedness.”

Edit this sentence to clarify and explain in more detail what features of the plasmids harboring carbapenemase genes you had characterized in this work.

RE: It was done.

Line 107:

“The complete nucleotide sequences of plasmids containing four carbapenemase genes, including blaKPC, blaNDM, blaOXA-48, and blaGES were retrieved from the GenBank database (https://www.ncbi.nlm.nih.gov/genbank/).”

I understand that your criterion of selection of plasmids was to have any of those 4 types of carbapenemase genes (blaKPC, blaNDM, blaOXA-48, and blaGES) not to have all the four types in a plasmid. It that is the case it is not clear in this sentence.

RE: Thank you for your comment. The correction was done.

Line 125:

oriTfnder tool (https://bioinfomml.sjtu.edu.cn/oriTf inder/)

Please correct the name of the tool and the url

RE: The correction has been done.

Line 147:

Indicate which is the plasmid with 5,803,733 bp

RE: The correction has been done.

Line 164:

“plasmids carrying four mentioned carbapenemase genes.”

This sentence is confusing because I understand that there are not 4 carbapenemase genes in each selected plasmid.

RE: Thank you for your comment. The correction has been done.

Line 276

“According to the MST results, STs predominated in each carbapenemase gene, including 276 blaKPC, blaNDM, blaOXA, and blaGES, were ST11, ST14, and ST12. Apart from the predominant 277 STs, some others, including ST37, ST35, ST16, ST392, ST147, ST17, ST23, ST101, ST307, 278 ST11, ST14, and ST437, were multi-harbor sequence types and associated with plasmids 279 containing three carbapenemase genes, including blaKPC, blaNDM, blaOXA.”

Please, add (Multilocus Sequence Typing) after MST and rewrite this sentence because its meaning is not clear.

RE: It was done.

Line 293

K.pneumonae -> K. pneumoniae

RE: The correction has been done.

In general please put K. pneumoniae in italics (it is not my case but microbiologists suffer a lot if you don’t)

RE: The correction has been done.

Line 320

“One of the most outstanding findings of the current study is the multi-harbor STs. STs 101, 320 147, and 16 are three important sequence types involved in the spread of K.pneumoniae isolates. Overall, the data from the current study showed that the plasmids associated with ST147 and ST101 had mostly IncL (with blaOXA) and IncFIB and IncFII replicon types (containing blaKPC and blaNDM), that were almost potentially conjugative or at least mobilizable. The predominant alleles were blaOXA-48, blaKPC-2, and blaNDM-1, however, blaOXA-10, blaOXA-181, blaOXA-232, blaNDM-5, blaNDM-7, blaNDM-29 and blaNDM-9 could also be found in plasmids associated with ST147”

This sentence needs to be rewritten.

RE: It was done.

Line 344

“…plasmids harboring the carbapenem/ genes.” → plasmids harboring the carbapenem resistance genes

RE: The correction has been done.

Line 361

“The co-existence of resistance genes belonging to different antibiotic clases,….” → The co-existence of different antibiotic classes resistance genes,….

RE: The correction has been done.

Line 365

“multi-harbor carbapenemases and international STs (e.g. ST11, ST23, ST14, and ST12) could highly affect the exacerbation of the antimicrobial resistance in K. pneumoniae.”

IMHO this sentence has not sense.

RE: It was re-written. 

Finally, I want to encourage the authors to continue working on it because I think they can do a better job on this topic, which is very important and needs a deep analysis.

---

## [Editor Report · Decision Letter 1]

18 Sep 2023

Decoding the genetic structure of conjugative plasmids in international clones of Klebsiella pneumoniae: A deep dive into blaKPC, blaNDM, blaOXA-48, and blaGES genes

PONE-D-23-18985R1

Dear Dr. Badmasti,

We’re pleased to inform you that your manuscript has been judged scientifically suitable for publication and will be formally accepted for publication once it meets all outstanding technical requirements.

Kind regards,

Farah Al-Marzooq, MD, PhD

Academic Editor

PLOS ONE

Additional Editor Comments (optional):

The manuscript was improved after revision
---

## [Editor Report · Acceptance letter]

7 Nov 2023

PONE-D-23-18985R1 

Decoding the genetic structure of conjugative plasmids in international clones of *Klebsiella pneumoniae*: A deep dive into *bla*_KPC_, *bla*_NDM_, *bla*_OXA-48_, and *bla*_GES_ genes 

Dear Dr. Badmasti:

I'm pleased to inform you that your manuscript has been deemed suitable for publication in PLOS ONE. Congratulations! Your manuscript is now with our production department. 

Kind regards, 

on behalf of

Dr. Farah Al-Marzooq 

Academic Editor

PLOS ONE